# Architecture and self-assembly of the jumbo bacteriophage nuclear shell

Thomas G. Laughlin[1,7], Amar Deep[2,7], Amy M. Prichard[1], Christian Seitz[3], Yajie Gu[2], Eray Enustun[1], Sergey Suslov[1], Kanika Khanna[1,6], Erica A. Birkholz[1], Emily Armbruster[1], J. Andrew McCammon[3,4], Rommie E. Amaro[3], Joe Pogliano[1✉], Kevin D. Corbett[2,3✉] & Elizabeth Villa[1,5✉]

Bacteria encode myriad defences that target the genomes of infecting bacteriophage, including restriction–modification and CRISPR–Cas systems[1]. In response, one family of large bacteriophages uses a nucleus-like compartment to protect its replicating genomes by excluding host defence factors[2–4]. However, the principal composition and structure of this compartment remain unknown. Here we find that the bacteriophage nuclear shell assembles primarily from one protein, which we name chimallin (ChmA). Combining cryo-electron tomography of nuclear shells in bacteriophage-infected cells and cryo-electron microscopy of a minimal chimallin compartment in vitro, we show that chimallin self-assembles as a flexible sheet into closed micrometre-scale compartments. The architecture and assembly dynamics of the chimallin shell suggest mechanisms for its nucleation and growth, and its role as a scaffold for phage-encoded factors mediating macromolecular transport, cytoskeletal interactions, and viral maturation.

Over billions of years of conflict with bacteriophages (phages), plasmids, and other mobile genetic elements, bacteria have evolved an array of defensive systems to target and destroy foreign nucleic acids[1]. In turn, phages have evolved mechanisms, including anti-restriction and anti-CRISPR proteins, that counter specific bacterial defence systems[5–7]. We recently showed that a family of 'jumbo phages'—named for their large genomes (typically over 200 kb) and large virion size (capsids approximately 145 nm in diameter with contractile tails around 200 nm in length)—assemble a selectively permeable, protein-based shell that encloses the replicating viral genome and is associated with a unique phage life cycle[2,8] (Fig. 1a). This micrometre-scale nucleus-like compartment, termed the 'phage nucleus', forms de novo upon infection and grows with the replicating viral DNA. Meanwhile, phage proteins are synthesized in the host cell cytoplasm. These include PhuZ, a phage-encoded tubulin homologue that assembles into filaments that treadmill to transport empty capsids from their assembly sites at the host cell membrane to the surface of the phage nucleus. Capsids dock to the phage nuclear shell and are filled with viral DNA before detaching and completing assembly with phage tails. Mature particles are released by host cell lysis. In contrast to other characterized anti-restriction systems, the phage nuclear shell renders jumbo phages broadly immune to DNA-targeting host restriction systems, including CRISPR–Cas, throughout infection by serving as a physical barrier between the viral DNA and host nucleases[3,4].

We previously showed that the phage nuclear shell incorporates at least one abundant phage-encoded protein[2,8], but the overall composition and architecture of this structure remain largely unknown. It is also unclear how these phages address the challenges arising from the separation of transcription and translation, specifically the need for directional transport of mRNA out of the phage nucleus and transport of DNA-processing enzymes into the phage nucleus[2]. Finally, how genomic DNA produced in the phage nucleus is packaged into capsids assembled in the cytosol is also unknown[9].

## Architecture of the phage nuclear shell

To gain insight into the native architecture of the phage nuclear shell, we performed focused ion-beam milling coupled with cryo-electron tomography (cryoFIB–ET) of *Pseudomonas chlororaphis* 200-B cells infected with jumbo phage 201phi2-1 (317 kb genome) at 50–60 min post-infection (mpi) (typical time to lysis is around 90 mpi) (Fig. 1 and Extended Data Fig. 1a–i). The observed phage nuclei were pleomorphic compartments devoid of ribosomes, bounded by a proteinaceous shell that is approximately 6 nm in thickness (Fig. 1b–d and Extended Data Fig. 1). Close inspection of the compartment perimeter revealed repeating doublets of globular densities with approximately 11.5 nm spacing, suggesting that the shell consists of a single layer of proteins in a repeating array (Fig. 1e and Extended Data Fig. 1j). Furthermore, we occasionally captured face-on views showing a square lattice of densities with the same repeat spacing, reinforcing the idea of a repeating array of protomers (Extended Data Fig. 1k).

Using subtomogram analysis to average over low-curvature regions from eight separate 201phi2-1 nuclei, we obtained a reconstruction with a resolution of about 24 Å of the phage nuclear shell (Fig. 1f,g, Extended Data Fig. 1h–o and Supplementary Table 1). The reconstruction reveals the shell as a quasi-*p*4, or square, lattice (quasi because

[1]Department of Molecular Biology, School of Biological Sciences, University of California San Diego, La Jolla, CA, USA. [2]Department of Cellular and Molecular Medicine, University of California San Diego, La Jolla, CA, USA. [3]Department of Chemistry and Biochemistry, University of California San Diego, La Jolla, CA, USA. [4]Department of Pharmacology, University of California San Diego, La Jolla, CA, USA. [5]Howard Hughes Medical Institute, University of California San Diego, La Jolla, CA, USA. [6]Present address: Department of Molecular and Cell Biology, University of California, Berkeley, CA, USA. [7]These authors contributed equally: Thomas G. Laughlin, Amar Deep. ✉e-mail: jpogliano@ucsd.edu; kcorbett@ucsd.edu; evilla@ucsd.edu

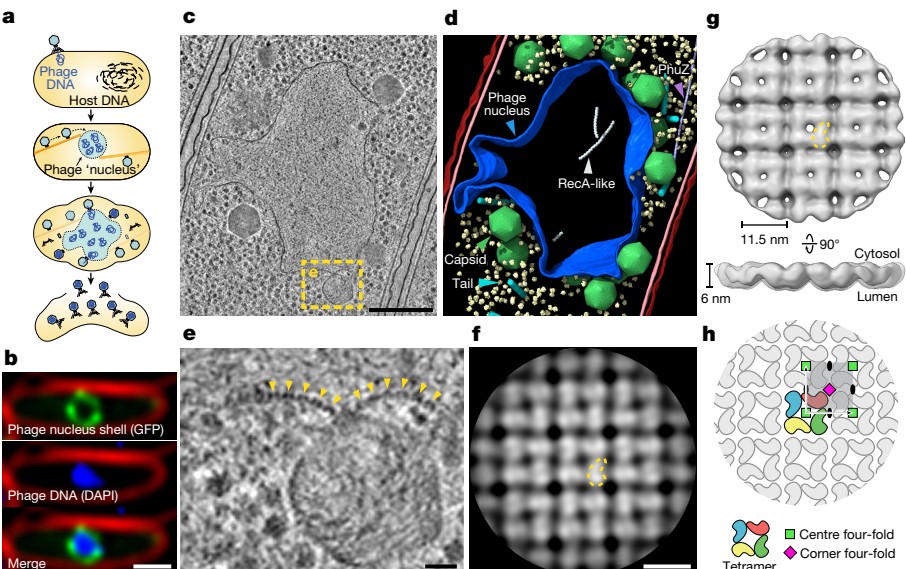

**Fig. 1 | In situ tomography and subtomogram analysis of the 201phi2-1 phage nucleus. a**, Schematic of the jumbo phage infection cycle. **b**, Fluorescence microscopy of a 201phi2-1-infected *P. chlororaphis* cell at 45 mpi (*n* = 5 independent experiments). Phage nucleus shell component gp105 (green) is tagged with GFP, phage DNA (blue) is stained with DAPI and the outer cell membrane (red) is stained with FM4-64. **c**, Tomographic slice of a phage nucleus in a 201phi2-1-infected *P. chlororaphis* cell at 50–60 mpi. **d**, Segmentation of the tomogram in **c**. Outer and inner bacterial membranes are shown in burgundy and pink, respectively. The phage nucleus is coloured blue. Phage capsids and tails are green and cyan, respectively. PhuZ and RecA-like protein filaments are light purple and white, respectively. A subset of 500 host ribosomes is shown in pale yellow. **e**, Enlarged view of the boxed region in **c**. Yellow arrows point to the repetitive feature of the phage nucleus perimeter. **f**, Slice of the cytosolic face of the subtomogram average of the repetitive feature in the phage nucleus perimeter with a comma-shaped subunit outlined in yellow. **g**, Cytosolic and side views of the shell subtomogram average isosurface with a single subunit outlined in yellow. **h**, Schematic representation of the *p442*-like arrangement of chimallin protomers. A 'centre' four-fold symmetry is indicated by a green square and a 'corner' four-fold symmetry is indicated by a magenta square. Scale bars: 1 μm (**b**), 250 nm (**c**), 25 nm (**e**) and 10 nm (**f**).

of variable curvature). The repeating unit is an 11.5 × 11.5 nm square tetramer approximately 6 nm thick, with internal four-fold rotational symmetry. These units form a square lattice with a second four-fold rotational symmetry axis at the corner of each square unit and two-fold axes on each side (the *p442* wallpaper group; Fig. 1h and Extended Data Fig. 1m,o). The four individual protomer densities within each 11.5 × 11.5 nm unit measure around 6 × 6 × 7.5 nm, dimensions consistent with a protein with a molecular mass of about 70 kDa. Thus, the phage nuclear shell appears to be predominantly composed of a single protein component arranged in a square lattice.

## Chimallin is the principal shell protein

We previously showed that the abundant and early-expressed 201phi2-1 protein gp105 becomes integrated into the nuclear shell[2]. Furthermore, 201phi2-1 gp105 has a molecular weight of 69.5 kDa (631 amino acids), consistent with the size of an individual protomer density from our in situ cryo-electron tomography (cryo-ET) map. Along with the high apparent compositional homogeneity of the shell as observed by cryo-ET, these data support 201phi2-1 gp105 as the principal component of the phage nuclear shell. Homologues of 201phi2-1 gp105 are encoded by a large set of jumbo phage that infect diverse bacteria including *Pseudomonas*, *Vibrio*, *Salmonella*, and *Escherichia coli*, but these proteins bear no detectable sequence homology to any other proteins. Because of its role in protecting the phage genome against host defences, we named this protein chimallin (ChmA) after the *chimalli*, a shield carried by ancient Aztec warriors[10].

To understand the structure and assembly mechanisms of chimallin, we expressed and purified 201phi2-1 chimallin from *E. coli*. Size-exclusion chromatography coupled to multi-angle light scattering (SEC–MALS) of purified chimallin indicated a mixture of oligomeric states including monomers, well-defined assemblies of approximately

1.2 MDa, and larger heterogeneous species ranging from 4 to 13 MDa (mean of 6.9 MDa; Fig. 2a). Cryo-ET of the largest species revealed pleomorphic, closed compartments with near-identical morphology to the phage nuclear shell that we observe in situ (Fig. 2b and Extended Data Fig. 2). Analysis of the smaller, more defined assemblies by cryo-ET revealed a near-homogeneous population of cubic assemblies with a diameter of around 22 nm, and a minor population of rectangular assemblies with dimensions of approximately 22 × 33.5 nm (Fig. 2c).

We next acquired cryo-electron microscopy (cryo-EM) data and performed single-particle analysis (SPA) of the defined chimallin assemblies (Extended Data Fig. 3). Two-dimensional class averages revealed that each cubic particle consists of six chimallin tetramers (24 protomers, 1.67 MDa) arranged to form a minimal closed compartment with apparent octahedral (*O*, 432) symmetry (Extended Data Fig. 3a). Similarly, the minor population of approximately 22 × 33.5 nm rectangular particles are assemblies of ten chimallin tetramers (40 protomers, 2.78 MDa) with apparent *D*4 symmetry (Extended Data Fig. 3g). Each tetrameric unit in these assemblies is an 11.5 × 11.5 nm square, in line with our in situ subtomogram analysis of the phage nuclear shell. Three-dimensional reconstruction of the cubic particles with enforced *O* symmetry resulted in an approximately 4.4 Å-resolution density map with distorted features (Fig. 2d), probably arising from the inherent plasticity of these assemblies breaking symmetry. Localized reconstruction of the square faces from each particle with *C*4 symmetry resulted in an improved density map at approximately 3.4 Å resolution. Further reduction of the structure to focus on an individual protomer resulted in the highest quality map at about 3.1 Å resolution, enabling atomic modelling of the chimallin protomer (Fig. 2e,f, Supplementary Video 1 and Supplementary Tables 2–4).

Chimallin folds into a compact two-domain core with extended N- and C-terminal segments (Fig. 2f,g). The N-terminal domain (residues 62–228) shows little structural homology to any characterized

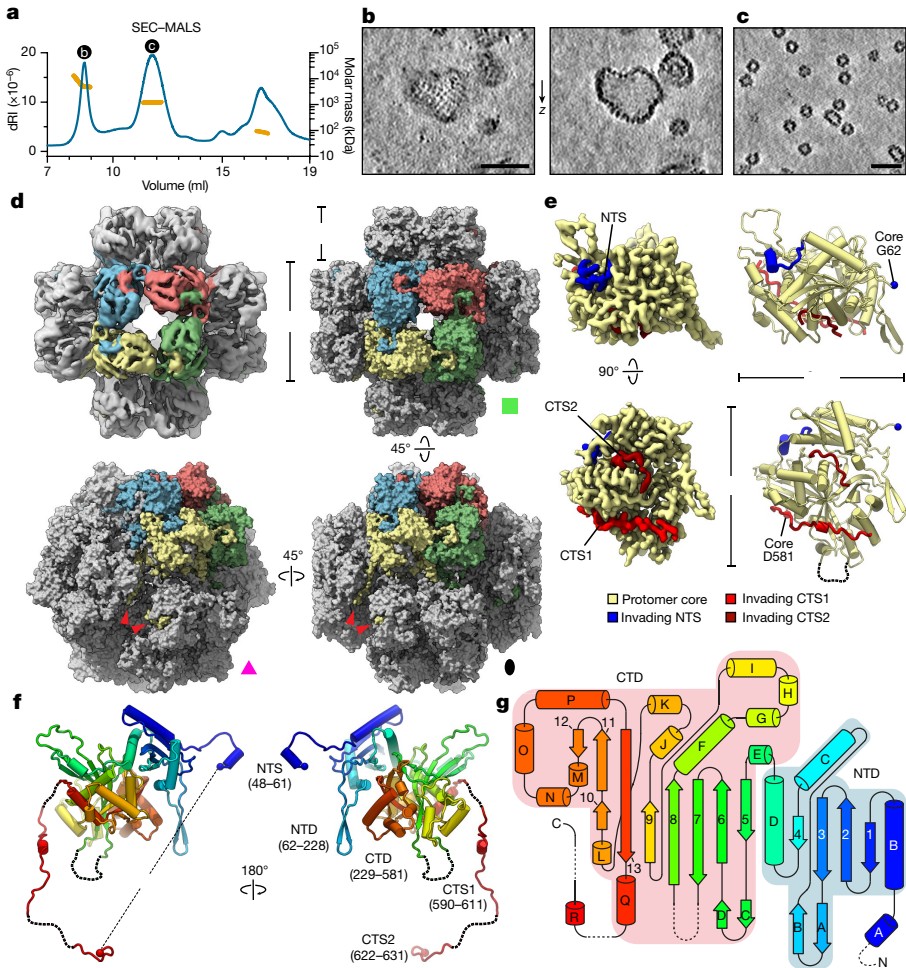

**Fig. 2 | In vitro cryo-EM structure of the 201phi2-1 phage nuclear shell protein chimallin. a**, SEC–MALS analysis of purified 201phi2-1 chimallin. The measured molar masses of the three peaks are 6.9 MDa (range 4–13 MDa), 1.2 MDa and 87 kDa (left to right). dRI, differential refractive index. See Extended Data Fig. 5 for molar mass measurements by SEC–MALS. **b,c**, Z-slices from tomograms of samples from the correspondingly labelled SEC–MALS peaks in **a**. The full field of view of **b** is provided in Extended Data Fig. 2. **d**, Top left, O-symmetrized reconstruction of the chimallin cubic assembly viewed along the four-fold axis. The protomers of one four-fold face are coloured. Top right, surface representation of the chimallin cubic assembly model viewed along the four-fold axis. Bottom right and bottom left, views of the model along the two-fold and three-fold axes, respectively. Red arrowheads point to the C-terminal segments of the yellow protomer. The green square, pink triangle and black oval indicate that the corresponding panels are viewed down the particle's four-fold, three-fold and two-fold rotational symmetry axes, respectively. **e**, Localized asymmetric reconstruction of the chimallin protomer (left) and cartoon model (right). Invading N- and C-terminal segments from neighbouring protomers are coloured blue (NTS), red (CTS1) and burgundy (CTS2). Resolved core protomer termini are shown as spheres. **f**, Rainbow-coloured cartoon model of the chimallin protomer conformation in the cubic assembly. Resolved N and C termini are shown as spheres. Domains and segments are labelled. Unresolved linkers are shown as dashed lines. **g**, A rainbow-coloured fold diagram of chimallin (blue at N terminus, red at C terminus) with α-helices labelled alphabetically and β-strands labelled numerically. The N- and C-terminal domains are highlighted in blue and red, respectively. Dashed lines indicate unresolved loops. Scale bars, 50 nm.

protein, adopting an αβ fold that is topologically similar only to an uncharacterized protein from *Enterococcus faecalis* (Extended Data Fig. 4a,b). The C-terminal domain (residues 229–581) adopts a GCN5-related *N*-acetyltransferase fold most similar to that of *E. coli* AtaT and related tRNA-acetylating toxins[11] (Protein Data Bank (PDB): 6AJM; root mean squared deviation (RMSD) 4.2 Å for 269 aligned Cα atom pairs) (Extended Data Fig. 4c–e). Although chimallin lacks the acetyltransferase active site residues of AtaT and related toxins, this structural similarity suggests that the jumbo phage nuclear shell may have evolved from a bacterial toxin–antitoxin system.

Atomic models of the *C*4-symmetric chimallin tetramer and quasi-*O*-symmetric cubic assemblies reveal the molecular basis for chimallin self-assembly. The chimallin protomer map contains three interacting peptide segments from the N and C termini of neighbouring protomers (Fig. 2d–f and Supplementary Video 1). Using the maps for the tetrameric face and full cubic assembly, we built models for the higher-order chimallin oligomers to understand the subunit interconnectivity. The N-terminal interacting segment (NTS: residues 48–61) of each protomer extends anticlockwise (as viewed from outside the cube) and docks against a neighbouring protomer's N-terminal domain within a given face of the cubic assembly, thus establishing intra-tetramer connections. Two extended segments of the C terminus (CTS1: residues 590–611; and CTS2: residues 622–631) establish inter-tetramer interactions. Although the linkers between the C-terminal domain and CTS1 (residues 582–589) and between CTS1 and CTS2 (residues 612–621) are unresolved in our maps, we could confidently infer the path of each protomer's C terminus within the cubic assembly. CTS1 extends from one protomer to a neighbouring protomer positioned anticlockwise around the three-fold symmetry axis of the cube, and CTS2 further extends anticlockwise to the third subunit around the same axis (Fig. 2d). Notably, the binding of CTS1 to the chimallin C-terminal domain resembles the interaction of the antitoxin AtaR with the AtaT

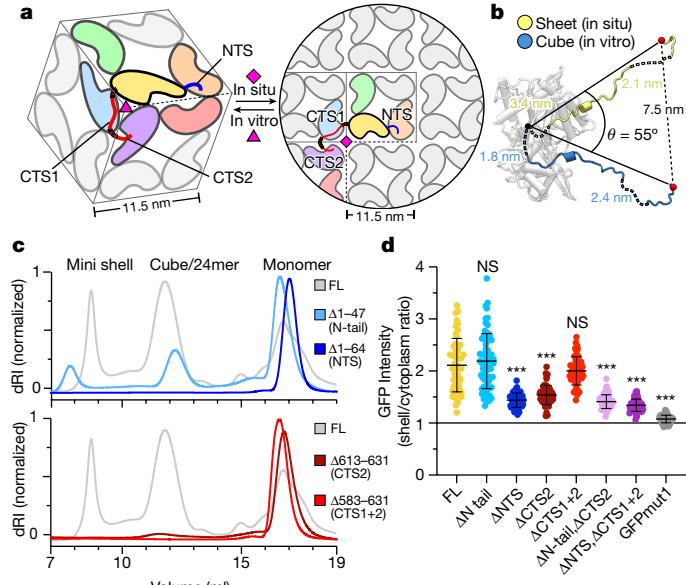

**a**, Protomer packing in the cubic 24mer assemblies (left) and flat sheet model (right). One protomer is coloured yellow with NTS in blue and CTS1/CTS2 in red. Protomers interacting directly with this focal protomer are coloured orange, green, blue, purple and red. Non-interfacing protomers are grey. Red dashed lines indicate locations of unresolved linkers, and pink symbols indicate 3- or 4-fold symmetry axes. **b**, Comparison of chimallin C terminus conformation in the in vitro sheet and the in situ cube. Distances spanned by each disordered segment (CTD–CTS1: residues 582–589 and CTS1–CTS2: residues 612–621) in the two models are noted. **c**, SEC–MALS of N- and C-terminal truncation mutants (ΔN tail, Δ1–47 (residues 48–631 are present); ΔNTS, Δ1–64 (residues 65–631 are present); ΔCTS2, Δ613–631 (residues 1–611 are present); ΔCTS1+2, Δ583–631 (residues 1–582 are present). See Extended Data Fig. 5 for molar mass measurements by SEC–MALS. **d**, Relative incorporation of eGFP–chimallin variants into the 201phi2-1 phage nucleus of infected *P. chlororaphis* cells. Incorporation is calculated as the ratio of GFP fluorescence per pixel in the shell versus outside the shell (details in Extended Data Fig. 6). Data are mean ± s.d. Unpaired *t*-test between a given variant and full-length (FL) (*n* = 67 cells); ***P* < 0.0001; ΔN tail: *n* = 51, *P* = 0.4131; ΔNTS: *n* = 53, *P* < 0.0001; ΔCTS1 (residues 1–611): *n* = 54, *P* < 0.0001; ΔCTS1+2: *n* = 50, *P* = 0.1884; ΔN tail, ΔCTS2 (residues 48–611): *n* = 58, *P* < 0.0001; ΔNTS, ΔCTS1+2 (residues 65–582): *n* = 63, *P* < 0.0001; eGFP: *n* = 60, *P* < 0.0001. The threshold for significance was Bonferroni-corrected to *P* < 0.007 to account for multiple hypothesis testing.

acetyltransferase toxin[11] (Extended Data Fig. 4e), further hinting that the phage nuclear shell could have evolved from a bacterial toxin–antitoxin system. Compared with the tightly packed chimallin tetramers mediated by the well-ordered NTS region, the length and flexibility of the linkers between the chimallin C-terminal domain, CTS1 and CTS2 suggest that flexible inter-tetramer packing enables chimallin to assemble into structures ranging from a flat sheet to the observed cubic assembly.

## Chimallin self-assembly and dynamics

To investigate the interconnectivity of chimallin protomers in the context of the phage nucleus, we docked copies of the high-resolution chimallin tetramer model into the in situ cryo-ET map. The chimallin tetramers fit well into the 24 Å-resolution cryo-ET map of the shell without clashes and with an overall map–model correlation coefficient of 0.56 (Extended Data Fig. 5a). To accommodate a flat sheet structure, the three-fold symmetry axis at each corner of the cubic assembly must be 'unfolded' into a four-fold symmetry axis (Fig. 3a and Supplementary Video 1). Since CTS1 and CTS2 mediate interactions across this

symmetry axis in the cubic assembly, this change requires that CTS1 rotate around 55° relative to the chimallin protomer core, and that CTS2 rotate the same amount relative to CTS1 (Fig. 3b). In the resulting sheet, each chimallin C terminus extends anticlockwise to contact the two neighbouring subunits in the new four-fold symmetry axis at the corners of the tetrameric units. The distances spanned by each disordered linker are similar in the flat sheet and the cubic assembly (Fig. 3b). Thus, both N- and C-terminal interacting segments contribute to shell self-assembly, with the C terminus in particular probably imparting significant structural plasticity to the phage nuclear shell while maintaining its overall integrity.

We next assessed the importance of the NTS and CTS regions for chimallin self-assembly both in vitro and in vivo. In vitro, deletion of either the NTS or CTS (CTS2 alone or both CTS1 and CTS2) completely disrupted self-assembly as measured by SEC–MALS (Fig. 3c and Extended Data Fig. 6). We expressed the same truncations in phage 201phi2-1-infected *P. chlororaphis* cells, and measured incorporation of eGFP–chimallin into the nuclear shell assembled by the phage-encoded full-length chimallin. We observed shell assembly in all cases, and found that deletion of the NTS, CTS or both partially compromised incorporation into the shell (Fig. 3d and Extended Data Fig. 7a). Consistent with this finding, expression of truncated chimallin did not strongly affect propagation of phage 201phi2-1 as measured by bacterial growth curves (Extended Data Fig. 7b). Overall, these data support the idea that chimallin NTS and CTS regions are important for efficient self-assembly of the chimallin shell, and show that self-assembly of full-length chimallin is robust even in the presence of chimallin protomers lacking these regions.

## Flexibility of the chimallin shell

The phage nucleus shields the viral genome in a manner similar to a viral capsid. However, unlike viral capsids, chimallin does not interact tightly with the encapsulated DNA. Indeed, estimation of the electrostatics of chimallin indicates that both the cytosolic and lumenal faces are negatively charged (Fig. 4a,b). The negative character of the phage nuclear shell probably reduces interactions with the enclosed DNA, thereby keeping the genetic material accessible for transcription, replication and capsid packaging.

Again, in contrast to viral capsids and other protein-based organelles such as bacterial microcompartments, which form regular assemblies with defined facets, the phage nuclear shell adopts a highly irregular morphology (Fig. 1c and Extended Data Fig. 1b–i). To identify the forms of conformational heterogeneity within the chimallin lattice, we performed an elastic network model analysis of a 3 × 3 tetramer chimallin sheet. This analysis indicated hinging not only along tetramer boundaries, which would lead to the cubic arrangement observed in vitro, but also within a given tetramer (Extended Data Fig. 8a–f and Supplementary Videos 2 and 3). Prompted by this analysis, we performed focused classification with our in situ data, which revealed three distinct classes of the central chimallin tetramer (Extended Data Figs. 1n,p and 8j). The predominant class shows a flat sheet, whereas two minor classes show the central unit raised (convex) or lowered (concave) by approximately 1 nm compared with surrounding units. Docking the tetramer model into maps representing concave, flat, and convex subpopulations indicated that the best fit is to the convex class (model–map correlation coefficient = 0.64), and the worst fit is to the concave class (model–map correlation coefficient = 0.51). Moreover, close inspection of the density representing a chimallin protomer in the different classes revealed that each protomer rotates approximately 25° between the concave and convex classes, with the inner side of the tetramer pinching inward in the convex class (Extended Data Fig. 8i,j). The tetramer model derived from the cubic assembly, which represents a highly convex state, shows a further approximately 25° inward tilt of each protomer. These observations suggest that the chimallin tetramer itself is flexible, with the C-terminal domains on the shell's inner face

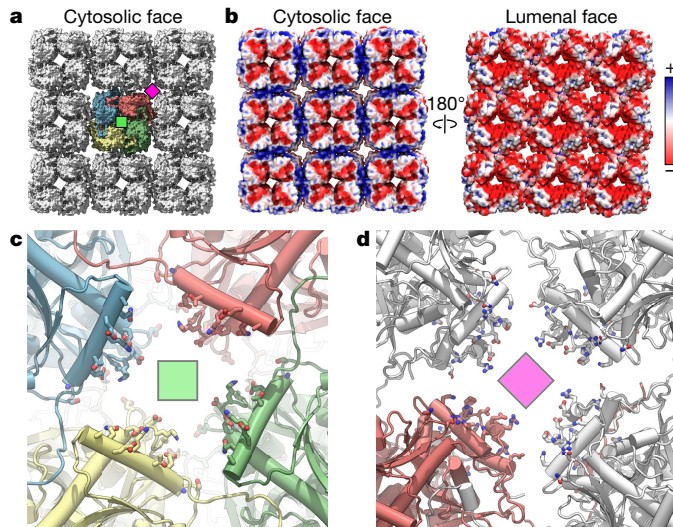

**a** Cytosolic face  **b** Cytosolic face   Lumenal face

**Fig. 4 | Electrostatics and pores of the chimallin shell. a**, Surface model of the 3 × 3 chimallin tetramer lattice viewed from the cytosol with the central tetramer coloured. A 'centre' four-fold symmetry is indicated by a green square and a 'corner' four-fold symmetry is indicated by a magenta square. **b**, Surface model of the cytosolic and lumenal faces of the chimallin lattice coloured by relative electrostatic potential (blue: positive; red: negative). **c**, Cytosolic views of the centre four-fold pore cartoon model with pore-facing residues (Supplementary Table 8) shown as sticks. The centre pores (*n* = 9) have an average volume of 798 ± 81 nm³ over the course of 300 ns molecular dynamics simulations (*n* = 5; Extended Data Fig. 8). **d**, Same as **c**, for the corner pore. The core pores (*n* = 4) have an average volume of 1,429 ± 227 nm³ over the course of simulations (*n* = 5 independent simulations; Extended Data Fig. 8).

rotating inward in convex regions of the shell, and outward in concave regions. Thus, the morphology and flexibility of the phage nuclear shell probably derive from both intra- and inter-tetramer motions.

To analyse the inter-tetramer motions in the in situ data beyond the subtomograms that we extracted from relatively flat regions of the nuclear shell, we examined the flexibility between chimallin tetramers in in situ nuclear shells by calculating the curvature of an annotated surface in a tomogram and estimating the angle between tetramers. Our analysis showed that chimallin inter-tetramer conformations can range from slightly concave—that is, −35° between neighbouring tetramers—to highly convex—that is, up to 80° between neighbouring tetramers, close to the maximum bend of 90° observed in the purified sample (Extended Data Fig. 8k,l). This enables the contorted shape observed in the cryo-ET data that presumably stems from addition of chimallin protomers to the lattice at multiple random locations, without distributing the strain throughout the lattice, rather than at a single privileged site.

## Narrow pores in the chimallin shell

Similar to the eukaryotic nucleus, the phage nucleus separates transcription and translation[2]. Thus, the phage nuclear shell must accommodate trafficking of mRNA out of the nucleus, and that of specific proteins into the nucleus. To determine whether the pores at the two four-fold symmetry axes of the chimallin lattice could serve as conduits for macromolecule transport, we used all-atom molecular dynamics to simulate the motions of a 3 × 3 flat sheet of chimallin tetramers (Fig. 4 and Supplementary Videos 4 and 5). Assessing the variability in these pores through five separate 300 ns simulations, we found that the restrictive diameters of both the centre and corner four-fold pores are around 1.4 nm on average, varying throughout the simulations from approximately 0 nm (that is, closed) to as wide as 2.3 nm (Extended Data Fig. 9 and Supplementary Table 6). These data strongly suggest

that the pores are too small to accommodate most folded proteins. However, this pore size is sufficient to enable exchange of metabolites, nucleotides and amino acids, and potentially large enough to support export of single-stranded mRNA molecules. A model for mRNA export is suggested by previous findings on viral capsid-resident RNA polymerases, which physically dock onto the inner face of the capsid and extrude mRNA co-transcriptionally through approximately 1.2-nm-wide pores[12,13]. Notably, chimallin-encoding jumbo phages have been shown to encode distinctive multi-subunit RNA polymerases, suggesting the co-evolution of chimallin and transcriptional machinery in this family[14]. Further study will be required to determine the protein(s) and sites responsible for protein import, as well as whether mRNA is extruded directly through the chimallin lattice pores.

## Nuclear shell architecture is conserved

The *E. coli* jumbo bacteriophage Goslar (237 kb genome) assembles a nuclear shell morphologically similar to those observed in the *Pseudomonas* phages 201phi2-1, PhiPA3 and PhiKZ[15]. Goslar encodes a divergent homologue of 201phi2-1 chimallin (gp189, 631 amino acids), with 19.3% overall sequence identity between the two proteins (Fig. 5a). We performed cryoFIB–ET on *E. coli* cells infected with Goslar at mid-infection (Methods), followed by subtomogram analysis of phage nuclei (Extended Data Fig. 10). The resulting approximately 30 Å-resolution reconstruction showed high overall similarity to the structure of the 201phi2-1 nuclear shell, with a square grid of 11.5 × 11.5 nm units (Fig. 5b and Extended Data Fig. 10h,i).

We next purified Goslar chimallin and characterized its self-assembly by SEC–MALS. Similar to 201phi2-1 chimallin, Goslar chimallin forms a mixture of monomers, defined assemblies of around 1.75 MDa, and large aggregates of around 10 MDa (Extended Data Fig. 11a). Cryo-EM analysis of the 1.75 MDa assemblies revealed cubic particles approximately 22 nm in diameter, paralleling those formed by 201phi2-1 chimallin (Fig. 5c and Extended Data Fig. 11b). We obtained a 4.2 Å-resolution structure of the overall Goslar chimallin assembly, and used a similar localized reconstruction procedure as for the 201phi2-1 chimallin to obtain a 2.6 Å-resolution structure of the Goslar chimallin tetramer, and a 2.3 Å-resolution structure of a single protomer (Fig. 5c,d and Extended Data Fig. 11c–g). Overall, Goslar chimallin shows high structural homology to 201phi2-1 chimallin despite the low overall sequence identity, with a Cα r.m.s.d. of 1.8 Å within a protomer and 4.8 Å over an entire *C*4-symmetric tetramer (Fig. 5e,f). The NTS, CTS1 and CTS2 segments of Goslar chimallin show near-identical interactions to neighbouring protomers compared with 201phi2-1 chimallin with the exception of CTS2, which is shorter in Goslar than in 201phi2-1 and shows a distinct set of interactions (Extended Data Fig. 5e–g and Supplementary Tables 8 and 9).

Recently, transmission electron micrographs of cryo-preserved lysates from *Salmonella* cells infected with the jumbo phage SPN3US revealed an unidentified square-lattice structure with a 13.5 nm periodicity[16]. We identified a diverged chimallin homologue (gp244) in SPN3US that shares only 10% identity with 201phi2-1 chimallin (Fig. 5a). The similar dimensions and overall morphology of the SPN3US lattice compared with the 201phi2-1 and Goslar nuclear shells strongly suggest that these structures are composed of chimallin. Further, during revision of this manuscript, a preprint reported a cryo-EM map of a chimallin sheet reconstituted in vitro from *Pseudomonas* jumbo phage PhiPA3[17] (gp53, 52.2% sequence identity to 201ph2-1 gp105). The PhiPA3 chimallin possesses a similar overall structure to 201phi2-1 and Goslar at the protomer level, but predominantly assembles in vitro as an oblique (*p*2) lattice in contrast to the predominantly square (*p*4) lattice observed in cells[17]. Thus, together with our findings on 201phi2-1 and Goslar, these data show that despite extremely low sequence conservation, chimallin proteins from divergent jumbo phages exhibit high structural conservation at both the level of an individual protomer and overall nuclear shell architecture.

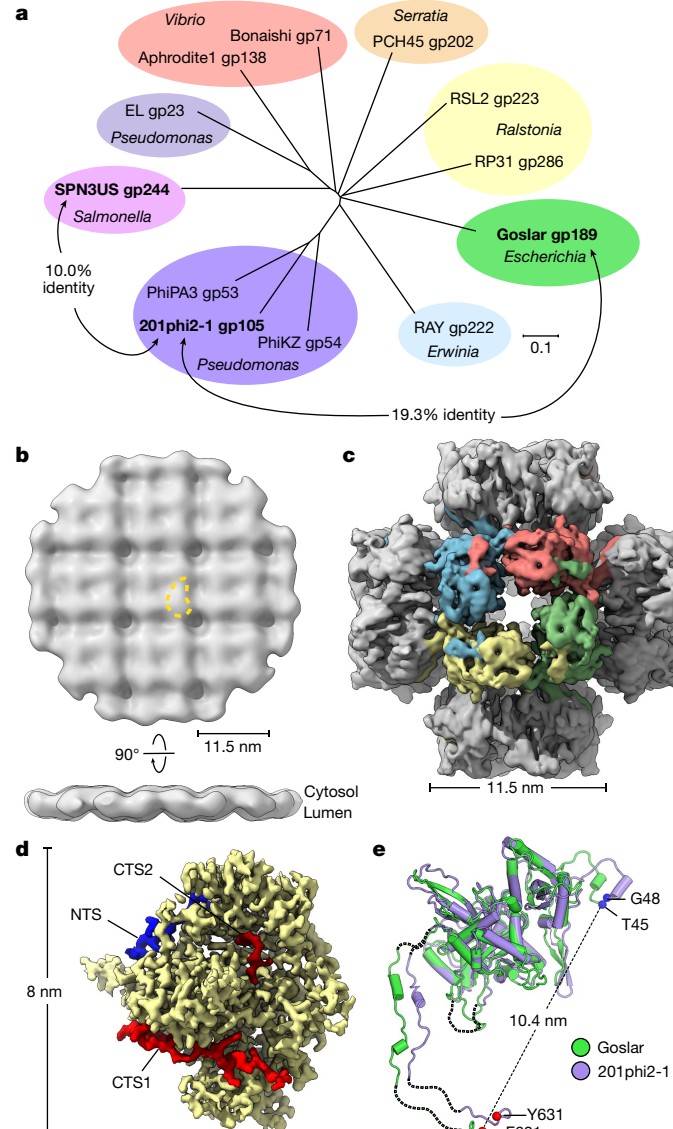

**Fig. 5 | Structural conservation of chimallin in the distantly related *E. coli* jumbo phage Goslar. a**, Unrooted phylogenetic tree of chimallin homologues. Homologues are listed as phage and gene product (gp) numbers (see Supplementary Table 7). Groups based on proximity are coloured and the host genus is indicated (Scale bar, 0.1 substitutions per position). **b**, In situ subtomogram reconstruction of the Goslar chimallin shell. A comma-shaped protomer is marked by a yellow dashed outline and cytosolic and lumenal faces are indicated. **c**, *O*-symmetrized map of the Goslar chimallin cubic assembly viewed along the four-fold axis. **d**, Localized asymmetric reconstruction of the Goslar chimallin protomer. Invading N- and C-terminal segments from neighbouring protomers are coloured blue (NTS), red (CTS1) and burgundy (CTS2). **e**, Superposition of the Goslar (green) and 201phi2-1 (purple) coordinate models for the cube confirmation of the protomers **e**. Resolved termini are shown as spheres for the protomers in **e**. The r.m.s.d. is 1.8 Å for the aligned protomers.

## Discussion

We have described the molecular architecture of the jumbo phage nuclear shell, a self-assembling, micrometre-scale proteinaceous compartment that segregates transcription and translation, largely excluding protein transport yet allowing selective protein import and mRNA export. We find that the shell is primarily composed of a single protein, chimallin, which self-assembles through its extended N and

C termini into a closed compartment. The nuclear shell is a square quasi-lattice that effectively balances integrity and flexibility, erecting a physical barrier between the replicating phage genome and host defences including restriction enzymes and CRISPR–Cas nucleases.

Chimallin is the first and most abundant protein produced upon phage 201phi2-1 infection of a host cell[2]. A key question is how chimallin specifically nucleates around the injected phage genome. Although it self-assembles in vitro at high concentration, chimallin does not form a phage nucleus by itself when overexpressed in uninfected cells[2,8], suggesting that the phage encodes one or more nucleation factors that promote the assembly of the shell around its own genome. These factors may be produced alongside chimallin in the infected cell or alternatively, may be injected along with the phage DNA upon initial infection. Thus, there is probably a window of time during the initial infection when the phage genome is not protected by the nuclear shell. It is not known how the phage protects itself during this period. We have consistently observed unidentified spherical bodies (USBs) in cells infected with either 201phi2-1 or Goslar, with internal densities consistent with tightly packed DNA (Extended Data Fig. 12). In 201phi2-1-infected cells, USBs have an average diameter of 201 nm (internal volume approximately $4.25 \times 10^{-3}$ μm$^3$), whereas in Goslar-infected cells, they have an average diameter of 182 nm (internal volume approximately $3.16 \times 10^{-3}$ μm$^3$). The internal volume of USBs in 201phi2-1-infected cells is 1.34 times that of those in Goslar-infected cells, closely matching the 1.33× ratio between the genome sizes of the two phages (317 kb for 201phi2-1 versus 237 kb for Goslar). For both 201phi2-1 and Goslar, the internal volume of USBs is 3.3–4.4 times the volume of their capsids, suggesting that if each USB contains one phage genome, the DNA is less densely packed in USBs compared with the capsid. Subtomogram analysis indicated that USBs do not have the same surface as the nuclear shell made of a single layer of chimallin; rather, the boundary's density is consistent with a lipid bilayer (Extended Data Fig. 12f–l). Notably, similarly sized compartments have been observed by thin-section transmission electron microscopy of jumbo phage PhiKZ and SPN3US infecting *Pseudomonas aeruginosa* and *Salmonella*, respectively[18,19]. Further work will be required to determine whether USBs represent unproductive infections that failed to pierce the inner membrane of the host, or instead represent a mechanism to protect the phage genome early in infection, before chimallin production and nuclear shell assembly.

Another unresolved question is how the phage nuclear shell grows concomitantly with phage genome replication. We propose that shell growth is accomplished by incorporating soluble chimallin subunits into the lattice through a presumably isoenthalpic transfer of the N- and C-terminal segments that bind the chimallin interfaces on the lattice to the new chimallin subunits, effectively breaking and subsequently resealing the existing lattice. The high propensity of chimallin to self-assemble, and the protein's abundance in infected cells, probably ensures that host defence factors have little opportunity to access the replicating phage genome during its growth.

We and others have shown that specific phage proteins are actively imported into the phage nucleus, whereas other proteins—including host defence nucleases—are excluded[3,4]. Given that the pores of the chimallin lattice are not large enough to allow passage of most folded proteins, these data suggest that the phage encodes minor shell components that mediate specific, directional transport of proteins and potentially phage-encoded mRNAs through the protein barrier. Similarly, specific proteins, either associated with or integrated into the shell, probably mediate interactions with the phage-encoded tubulin homologue PhuZ to position the phage nucleus[2] and enable phage capsids to dock on the shell surface for genome packaging[9]. Of note, the prohead protease of the jumbo phage PhiKZ has been shown to cleave chimallin (gp54) between its C-terminal domain and CTS1[20] (Supplementary Fig. 1), suggesting that proteolytic processing of chimallin may contribute to capsid docking and filling by locally disrupting lattice integrity.

The jumbo phage nucleus is a key example of convergent evolution to solve a problem—isolation of a genome from the surrounding cell contents—that was previously thought to have evolved only once in the history of life. In this Article, we have shown how the phage-encoded chimallin protein self-assembles into an effective nuclear–cytosolic barrier. This work sets the stage for future identification of minor shell components that manage shell nucleation and growth, mediate nuclear-cytoplasmic transport, and direct key steps in the phage life cycle including cytoskeletal interactions and genome packaging. Finally, the structural elucidation of the principal component of the phage nuclear shell opens the possibility of designing engineered protein-based compartments with sophisticated functions that span nanometre to micrometre scales.

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

## Methods

### Bacterial and phage growth conditions

*P. chlororaphis* 200-B[21] was grown on hard agar (HA) medium (10 g l$^{-1}$ Bacto-Tryptone, 5 g l$^{-1}$ NaCl and 10 g l$^{-1}$ agar in distilled water) at 30 °C. 201phi2-1 lysates were collected as previously described with minor modifications[2]. In brief, 0.5 ml from a dense *P. chlororaphis* culture grown in HA liquid media (HA medium with no agar added) was infected with 10 µl serial dilutions of high-titre 201phi2-1 lysate (10$^{11}$–10$^{12}$ pfu ml$^{-1}$), incubated for 15 min at room temperature, mixed with 4.5 ml of HA 0.35% top agar, poured over HA plates, and incubated overnight at 30 °C. Then, 5 ml phage buffer was added to web lysis plates the following day and incubated at room temperature for 5 h. The phage lysate was collected by aspiration, cell debris was pelleted by centrifugation at 3,220*g* for 10 min, and the resulting clarified phage lysate was stored at 4 °C. *E. coli* strain APEC 2248 was obtained from DSMZ and grown in LB medium at 37 °C. Goslar lysate (10$^{10}$ pfu ml$^{-1}$) was provided by J. Wittmann and stored at 4 °C. All bacterial strains used in this study are listed in Supplementary Table 10.

### Plasmid construction and expression

The pHERD-30T plasmid was used for expressing eGFP-tagged full-length and truncated Chimallin in *P. chlororaphis*[22,23]. All constructs were designed with eGFP fused to the N terminus and synthesized by GenScript. The plasmids were electroporated into *P. chlororaphis*, and 25 µg ml$^{-1}$ gentamicin sulfate was used for selection of colonies. All plasmids used in this study are listed in Supplementary Table 10.

### Live-cell fluorescence microscopy and image analysis

*P. chlororaphis* cells (5 µl of culture at $A_{600}$ = 0.6) were inoculated on imaging pads in welled microscope slides. The imaging pads were composed of 1% agarose, 25% HA broth, 25 µg ml$^{-1}$ gentamicin sulfate, 2 µg ml$^{-1}$ FM4-64, 0.2 µg ml$^{-1}$ DAPI and 1% arabinose to induce expression. The slides were incubated at 30 °C for 3 h in a humid chamber, and 10 µl of undiluted high-titre 201phi2-1 lysate was added to the pads to infect the cells 60 min before imaging. Samples were imaged using the DeltaVision Elite deconvolution microscope (Applied Precision). Images were deconvolved using the aggressive algorithm in the DeltaVision softWoRx program-v6.5.2.

All image analysis was performed on images prior to deconvolution. Average protein incorporation into the 201phi2-1 phage nucleus structure was determined using FIJI by measuring the mean grey value of the ring of GFP intensity that denotes the phage nucleus and the mean grey value of cytoplasmic GFP outside of this ring. The ratio of mean grey values of this ring GFP to cytoplasmic GFP was calculated as the average incorporation. Representative cells were chosen from each dataset, and a 3D graph of normalized GFP intensity was generated in MATLAB 2019a. Statistical analyses were performed using Prism-v9.3 (GraphPad Software).

### Growth curves

Culture of *P. chlororaphis* transformed with either an empty vector or one of the chimallin constructs were grown in HA broth containing 25 µg ml$^{-1}$ gentamicin sulfate to an $A_{600}$ between 0.6 and 0.8 and then back-diluted to an OD$_{600}$ of 0.1 in medium containing 1% arabinose in 96-well plates. Serial tenfold dilutions of 201phi2-1 lysate were added and growth monitored by $A_{600}$ at 10-min intervals for 8 h with continuous shaking at 30 °C. All growth curves were performed in duplicate, with duplicate wells (a total of four wells averaged per data point).

### Cryo-EM of in situ samples and image acquisition

For grid preparation of 201phi2-1 infections, host bacterial cells were infected on agarose pads as previously described[2] for 50–60 min. For grid preparation of Goslar infections, 10 agarose pads (1% agarose, 25% LB) were prepared in welled slides and spotted with 10 µl *E. coli* (APEC 2248) cells at an $A_{600}$ of ~0.35 then incubated at 37 °C for 1.5 h in a humidor. Ten microlitres of Goslar lysate from the DSMZ was added to each pad. At approximately 30 mpi, a portion of the infected cells were collected at room temperature and delivered for plunge-freezing. Infected cells were collected by the addition of 25 µl of 25% LB to each pad and gentle scraping with the bottom of an eppendorf tube followed by aspiration. A portion of the collection was aliquoted, the remainder was centrifuged at 6,000*g* for 45 s, resuspended with 0.25× volume of the supernatant, and a portion of that was diluted 1:1 in supernatant. The remaining cells incubated on pads at 37 °C until 90 mpi, at which point they were assessed for productive infections by light-microscopy. Plunging of samples began 20–30 min after removal from 37 °C, which significantly slows infection progression. Since phage nuclei were observed in this sample after cryoFIB-ET, the sample was suitable for the analyses performed in this study.

A volume of 4–7 µl of cells were deposited on R2/1 Cu 200 grids (Quantifoil) that had been glow-discharged for 1 min at 0.19 mbar and 20 mA in a PELCO easiGlow device shortly before use. Grids were mounted in a custom-built manual plunging device (Max Planck Institute of Biochemistry) and excess liquid blotted with filter paper (Whatman no. 1) from the backside of the grid for 5–7 s prior to freezing in a 50:50 ethane:propane mixture (Airgas) cooled by liquid nitrogen.

Grids were mounted into modified Autogrids (Thermo Fisher Scientific) compatible with cryo-focus ion-beam milling. Samples were loaded into an Aquilos 2 cryo-focused ion-beam/scanning electron microscope (TFS) and milled to generate lamellae approximately ~150–250 nm thick as previously described[24].

Lamellae were imaged using a Titan Krios G3 transmission electron microscope (Thermo Fisher Scientific) operated at 300 kV configured for fringe-free illumination and equipped with a K2-directed electron detector (Gatan) mounted post Quantum 968 LS imaging filter (Gatan). The microscope was operated in EFTEM mode with a slit-width of 20 eV and using a 70 µm objective aperture. Automated data acquisition was performed using SerialEM-v3.8b11[25] and all images were collected using the K2 in counting mode.

For lamellae of 201phi2-1-infected *P. chlororaphis*, tilt series were acquired at a 3.46 Å pixel size over a nominal range of ±51° in 3° steps with a grouping 2 using a dose-symmetric scheme[26] with a per-tilt fluence of 1.8 e$^-$ Å$^{-2}$ and total of about 120 e$^-$ Å$^{-2}$ per tilt series. Nine tilt series were acquired with a realized defocus range of −4.5 to −6 µm along the tilt axis. An additional two datasets of six tilt series each were collected at a pixel size of 4.27 Å with nominal tilt ranges of ±50° and ±60° in 2° steps with a grouping 2 using a dose-symmetric scheme with a per-tilt fluence of 1.8–2.0 e$^-$ Å$^{-2}$ and total of about 100–110 e$^-$ Å$^{-2}$ per tilt series.

For lamellae of Goslar-infected APEC 2248, tilt series were acquired at a 4.27 Å pixel size over a nominal range of ±56° in 2° steps with a grouping 2 using a dose-symmetric scheme with a per-tilt fluence of 2.6 e$^-$ Å$^{-2}$ and total of about 150 e$^-$ Å$^{-2}$ per tilt series. Twenty-one tilt series were acquired with a realized defocus range of −5 to −6 µm along the tilt axis.

### Image processing and subtomogram analysis of in situ cryo-EM data of the phage nucleus

All tilt series pre-processing was performed using Warp-v1.09 unless otherwise specified[27]. Tilt movies were corrected for whole-frame motion and aligned via patch tracking using Etomo (IMOD-v4.10.28)[28]. Tomograms were reconstructed with the deconvolution filter for visualization and manual picking in 3dmod (IMOD-v4.10.28). All subsequently reported resolution estimates are based on the 0.143-cutoff criterion of the Fourier shell correlations between masked, independently refined half-maps using high-resolution noise substitution to mitigate masking artefacts[29].

First, for the 201phi2-1-infected *P. chlororaphis* dataset collected at 3.46 Å per pixel, subtomogram averaging of the *P. chlororaphis* host cell ribosomes was performed in order to improve initial tilt series alignments using the recently developed multi-particle framework, M[30] (Extended Data Fig. 1n). A set of 400 particles were manually

picked across the tomograms, extracted at 20 Å per pixel, and aligned in RELION-v3.1.1 to generate an initial reference[31,32]. This data-derived reference was used for template-matching against 20 Å per pixel tomograms at a sampling rate of 15°. Template-matched hits were curated in Cube (https://github.com/dtegunov/cube), ultimately resulting in 17,169 particle picks. The initial particle set was extracted at 10 Å per pixel and subjected to Class3D, after which 11,148 particles were selected for further analysis. Refine3D of the curated particle set reached the binned Nyquist limit of 20 Å. The refined particles were imported into M-v1.09 at 3.46 Å per pixel. Three iterations of refinement were performed starting with image-warp and particle poses, then incorporating refinement of stage angles and volume-warp, and finally including individual tilt-movie alignment. This procedure resulted in a ribosome reconstruction at an estimated resolution of about 11 Å. Further refinement of the particles in RELION yielded a reconstruction at an estimated resolution of about 10 Å. Neither additional attempts at 3D classification nor multi-particle refinement leads to an improved ribosome reconstruction.

New tomograms were reconstructed at 20 Å per pixel using the ribosome alignment metadata. The perimeters of the 201phi2-1 phage nuclei were coarsely traced in these updated tomograms using 3dmod[32]. Traces were converted into surface models using custom MATLAB (MathWorks, v2019a) scripts and built-in Dynamo-v1.1.514 functions[33]. Points were sampled every 4 nm along the surface models, oriented normal to the surface (that is, positive $Z$ towards the cytosol), and extracted from normalized, contrast transfer function (CTF)-corrected tomograms at 10 Å per pixel in a 480 Å side-length box using Dynamo[33]. Initial orientations of the 66,887 extracted particles were curated using the Place Object plugin[34] for UCSF Chimera-v1.15[35] and incorrectly oriented particles were manually flipped.

To generate an initial reference, a subset of 17,622 particles from 2 tomograms were subjected to reference-free alignment in Dynamo-v1.1.514 for several iterations. For this procedure, no point-group symmetry was enforced, alignment was limited to 40 Å by an ad hoc lowpass filter each interaction, and the out-of-plane searches were restricted to prevent flipping of sidedness. The alignment converged to yield a reconstruction conforming to an apparent square ($p4$, 442) lattice. Analysis of particle positions and orientations using Place Objects[34] and neighbour plots[36] were consistent with reconstructed average and indicated a spacing between approximately 11.5 nm between congruent four-fold axes.

The initial reference was subsequently used to align the entire dataset for a single iteration in Dynamo. For this step, alignment was limited to 40 Å, the out-of-plane searches were restricted to prevent flipping of sidedness, $C4$ symmetry enforced, and a box-wide by 240 Å cylindrical alignment mask applied. Inspection of particle positions and orientations using Place Objects[34] and neighbour plots[36] were again consistent with a square-lattice-like arrangement. To deal with the initial over-sampling, particle duplicates were identified as those within 9 nm centre-to-centre distance of another particle and the one with the lower cross-correlation to the reconstruction removed, which resulted in 21,165 retained particles. In addition, a geometry-based cleaning step was performed to remove particles with less than three neighbours within 10 to 13 nm, which resulted in 8,454 retained particles.

The curated particle set was split into approximately equal half-sets on a per-tomogram basis, converted to the STAR file format using the dynamo2m-v0.2.2 package[37], and re-extracted into a 480 Å-side-length box at 5 Å per pixel in Warp for use in RELION. A round of Refine3D was performed using a 40 Å lowpass filtered reference, $C4$ symmetry, local-searches starting at 3.7°, and a box-wide soft shape mask. This resulted in a reconstruction at an estimated resolution of 24 Å for the 8,454-particle set.

Classification without alignment was performed using a 320 Å spherical mask and $C4$ symmetry, to promote convergence from the relatively low particle count. This differentiated three distinct classes corresponding to concave (2,033 particles), flat (4,475 particles) and convex (945 particles) states of the central tetramer, along with a noisy class (1,001 particles). The particles corresponding to the interpretable classes were re-extracted into a 320 Å box at 5 Å per pixel and subjected to 3D auto-refinement as described for the consensus reconstruction. The estimated resolutions for the lowered, intermediate, and raised classes were 20 Å, 18 Å and 23 Å, respectively. Refinement in M of either the consensus particle set or the three aforementioned classes above did not yield notable improvements in the reconstructions. This may be attributed to prior refinement of the tilt series alignment, the most resolution-limiting factor, using the host ribosomes[30].

The 12 tilt series of 201phi2-1-infected *P. chlororaphis* dataset collected at 4.27 Å per pixel were pre-processed and host ribosomes averaged as described above. The ribosome reconstruction from above was lowpass filtered to 40 Å and used for template-matching in Warp-v1.09, and curated in Cube to yield 47,469 particle positions. Ribosomes were extracted at 10 Å per pixel and subjected to reference-free 3D classification in RELION-v3.1.1 which resulted in 15,782 subtomograms. Masked auto-refinement resulted in a reconstruction with an estimated resolution of 28 Å. Refinement of tilt series parameters in M improved the resolution to about 20 Å (not shown).

For the 201phi2-1 nucleus in the 4.27 Å per pixel dataset, we were unable to completely resolve the quasi-lattice register using either an ab initio reference or the reconstruction from above as determined by neighbour plots. Thus, this dataset was solely included in the analysis of the unidentified spherical bodies and not further subtomogram averaging.

The tilt series of Goslar-infected APEC 2248 were pre-processed and host ribosomes averaged similarly as described above. An initial host ribosome reference was generated from 400 randomly selected particles and used for template-matching in Warp-v1.09, and curated in Cube to yield 98,981 particle positions. Ribosomes were extracted at 10 Å per pixel and subjected to reference-free 3D classification in RELION-v3.1.1 which distinguished 70S and 50S classes containing 46,056 and 3,710 particles, respectively. Particles were re-extracted at 6 Å per pixel and subjected to 3D auto-refinement to yield 20 Å and 14 Å for the 50S and 70S classes, respectively. Refinement of tilt series parameters in M, followed by an additional round of 3D auto-refinement at 4.27 Å per pixel resulted in 12 Å and 8.54 Å (Nyquist limit of the data) for the 50S and 70S, respectively.

For the Goslar nucleus, the nuclei perimeters were traced from 6 tomograms and used to extract over-sampled points normal to the surface, which resulted in 42,416 initial particles. A subset of 10,512 particles were used to generate an ab initio reference in $C1$ as described above. The initial Goslar reference presented a similar spacing (~11.5 nm) and apparent $C4$ symmetry as the 201phi2-1 reconstruction, however the average converged on the opposite four-fold axis. For ease of subsequent analysis, the centre of the Goslar initial reference was shifted to match 201phi2-1. Alignment of the entire dataset was performed as described above and distance-based cleaning post-alignment resulted in 4,501 particles for further processing. Alignment and reconstruction in RELION enforcing $C4$ symmetry resulted in a reconstruction with an estimated resolution of 27 Å. Similar to the 201phi2-1, 3D classification of the consensus refinement without alignment separated the data into classes in which the central tetramer appeared concave (2,802) or convex (1,699). Refinement of these classes resulted in reconstruction with estimated resolutions of 20 Å and 25 Å for the lowered and raised classes, respectively.

### Analysis of unidentified spherical bodies

USBs were manually identified and their maximal apparent diameters measured from their line intensity profiles in 20 Å per pixel tomograms using FIJI[38]. In order to assess whether the surfaces of these compartments possessed an underlying structure, we attempted subtomogram analysis of the compartment surfaces essentially as described in[37,39]. We were unable to obtain a reconstruction exhibiting a regular underlying structure as assessed both visually and by neighbour plots. However, despite their differing exterior membrane, the interior density of the USBs is visually consistent with nucleic acid like that of the interior of the phage nucleus.

## Segmentation and visualization of in situ tomography data

Segmentation of host cell membranes and the phage nucleus perimeter was performed on 20 Å per pixel tomograms by first coarsely segmenting using TomoSegMemTV[40] followed by manual patching with Amira-v6.7 (TFS). For the purposes of segmentation, phage capsids, tails, PhuZ, and RecA-like particles were subjected to a coarse subtomogram averaging procedure using particles sampled at 20 Å per pixel. For capsids, all particles were manually picked. Reference-generation and alignment of capsids was performed by enforcing icosahedral symmetry with Relion-v3.1.1[31,32] (despite the capsids possessing C5 symmetry) in order to promote convergence from the low number of particles. For the phage tails, the start and end points along the filament axis were defined manually and used to seed over-sampled filament models in Dynamo-v1.1.514[33,41]. An initial reference for the tail was generated using Dynamo-v1.1.514 from two full-length tails with clear polarity. The resulting reference displayed apparent C6 symmetry, which was enforced for the alignment of all tails from a given tomogram using Dynamo-v1.1.514 and Relion-v3.1.1. Similar to the phage tails, the PhuZ and RecA-like filaments were picked and refined but without enforcing symmetry. We do not report resolution claims for these averages and use them solely for display purposes in segmentations. Duplicate particles were removed and final averages were placed back in the reference-frame of their respective tomograms using dynamo_table_place. For clarity, a random subset of 500 ribosomes were selected for display in the segmentation.

## Surface curvature estimates from segmentation

The segmentation of the phage nucleus, depicted in Fig. 1d, sampled at 2 nm per pixels, was used to estimate the principal curvature of the shell using PyCurv[42]. PyCurv was run with default parameters and a hit radius of three pixels. For visualization purposes, the principal curvature values ($\kappa_1$, $\kappa_2$) were converted from a radius of curvature ($r$ (nm$^{-1}$)) to an angle ($\theta$ (degrees)) using the following formula:

$$\theta = 2\arctan\left(\frac{s}{2r}\right)$$

where $r$ is the radius of curvature (inverse of $\kappa_n$) and $s$ is the side length of the polygon circumscribed, taken as 5.75 nm.

## Protein expression and purification

Full-length Chimallin from bacteriophages 201phi2-1 (gp105; NCBI Accession YP_001956829.1) and Goslar (gp189; NCBI Accession YP_009820873.1) were cloned with an N-terminal TEV protease-cleavable His$_6$ tag using UC Berkeley Macrolab vector 2-BT (Addgene #29666). Truncations and other modified constructs were cloned by PCR mutagenesis and isothermal assembly, and inserted into the same vector. Proteins were expressed in *E. coli* Rosetta2 pLysS (Novagen) by growing cells to $A_{600} = 0.8$, inducing expression with 0.3 mM IPTG, then growing cells at 20 °C for 16–18 h. Cells were harvested by centrifugation and resuspended in buffer A (50 mM Tris pH 7.5, 10 mM imidazole, 300 mM NaCl, 10% glycerol, and 2 mM β-mercaptoethanol), then lysed by sonication and the lysate cleared by centrifugation. Protein was purified by Ni$^{2+}$ affinity method. The purified proteins were centrifuged briefly to settle down the floating particles (visible large assemblies and aggregated proteins). The proteins were dialysed into buffer B (20 mM Tris pH 7.5, 250 mM NaCl, 2 mM β-mercaptoethanol) and the N-terminal histidine tag was cleaved using TEV protease with overnight incubation at 4 °C. The retrieved tagless proteins were further purified for homogeneity through Superose 6 Increase 10/300 GL column (Cytiva) in buffer B. The quality of purified proteins was verified by SDS–PAGE analysis.

For analysis by SEC–MALS, a 100 µl sample of protein at 2 mg ml$^{-1}$ was passed over a Superose 6 Increase 10/300 GL column (Cytiva) in buffer B. Light scattering and refractive index profiles were collected by miniDAWN TREOS and Optilab T-rEX detectors (Wyatt Technology), respectively, and molecular weight was calculated using ASTRA v. 8 software (Wyatt Technology).

## Cryo-EM of in vitro samples and image acquisition

For grid preparation, freshly purified recombinant 201phi2-1 chimallin was collected from size-exclusion chromatography (estimated concentration of 4 µM of the monomer, 0.3 mg ml$^{-1}$). Immediately prior to use, R2/2 Cu 300 grids (Quantifoil) were glow-discharged for 1 min at 0.19 mbar and 20 mA in a PELCO easiGlow device. Sample was applied to a grid as a 3.2 µl drop in the environmental chamber of a Vitrobot Mark IV (Thermo Fisher Scientific) held at 16 °C and 100% humidity. Upon application of the sample, the grid was blotted immediately with filter paper for 3 s prior to plunging into a 50:50 ethane:propane mixture cooled by liquid nitrogen. Grids were mounted into standard AutoGrids (Thermo Fisher Scientific) for imaging. Grids for recombinant Goslar chimallin protein were prepared similarly, but with the modification that the sample was concentrated to approximately 33 µM of the monomer (2.5 mg ml$^{-1}$) prior to plunge-freezing. The void peaks from each purification were frozen similarly at the eluted concentration after dilution 1:1 with 6 nm BSA-tracer gold (Electron Microscopy Sciences).

All samples were imaged using a Titan Krios G3 transmission electron microscope (Thermo Fisher Scientific) operated at 300 kV configured for fringe-free illumination and equipped with a K2-directed electron detector (Gatan) mounted post Quantum 968 LS imaging filter (Gatan). The microscope was operated in EFTEM mode with a slit-width of 20 eV and using a 70 µm objective aperture. Automated data acquisition was performed using SerialEM-v3.8b11[25] and all images were collected using the K2 in counting mode.

For the 201phi2-1 24mer sample, tilt series were acquired using a pixel size of 1.376 Å with a per-tilt fluence of 4.7 e$^-$ Å$^{-2}$ using a dose-symmetric scheme[25] from ±51° in 3° steps and a grouping 3, resulting in a fluence of 164.5 e$^-$ Å$^{-2}$ per tilt series. In total 4 tilt series were collected with a realized defocus of −2.5 to −4 µm along the tilt axis. Movies for SPA were recorded at a pixel size of 1.075 Å with fluence of 42.6 e$^-$ Å$^{-2}$ distributed uniformly over 40 frames. Automated data acquisition was performed using image shift with active beam tilt compensation to acquire nine movies per hole per stage movement. In total 4,192 movies were acquired with a realized defocus range of −0.1 to −1.5 µm.

For the Goslar 24mer sample, movies for SPA were recorded at a pixel size of 0.8452 Å with fluence of 40 e$^-$ Å$^{-2}$ distributed uniformly over 44 frames. Automated data acquisition was again performed using image shift with active beam tilt compensation to acquire 10 movies per hole per stage movement. In total, 3921 movies were acquired with a realized defocus range of −0.1 to −1.5 µm.

For void peak samples, tilt series were acquired similarly to that of the 201phi2-1 24mer sample but using a pixel size of 1.752 Å and tilt-range of ±60°.

## Image processing of in vitro cryo-EM data

All movie pre-processing was performed using Warp-v1.09 unless otherwise specified[27]. Tilt-movies of the 201phi2-1 chimallin were corrected for whole-frame motion and aligned via patch tracking using Etomo (IMOD-v4.10.28)[28]. Tomograms were reconstructed with the deconvolution filter for visualization and manual picking of subtomograms using 3dmod (IMOD-v4.10.28)[43]. A total of 203 manually picked subtomograms and their corresponding 3D CTF volumes were reconstructed with a 288 Å side length. Subtomograms were aligned and averaged initially in *C*1 by reference-free refinement as implemented in RELION-v3.1.1[31,32] to an estimated resolution of 22 Å. The *C*1 reconstruction displayed features consistent with a cubic assembly of the 201phi2-1 chimallin protomers. Thus, an additional round of refinement using the *C*1 reconstruction as a reference and enforcing *O* point-group symmetry improved the estimated resolution to 18 Å.

For the single-particle 201phi2-1 chimallin data, movies were motion-corrected with exposure-weighting and initial CTF parameters

estimated using 5 × 5 grids. Micrographs were culled by thresholding for an estimated defocus in the range of 0.3–1.5 μm and CTF-fit resolutions better than 6 Å resulting in 4,098 micrographs for further processing. An initial set of 140,782 particle positions were picked with BoxNet2 (Warp-v1.09)[27] using a model re-trained on 20 manually curated micrographs and using a threshold of 0.95. Particle images were extracted using a 396 Å side length. All further processing was performed using RELION-v3.1.1[32] unless otherwise specified. A single round of reference-free 2D-classification was performed and the 128,798 particle images assigned to the averages displaying internal features were selected for further processing. At this stage, analysis of the 2D averages suggested the presence of four-fold, three-fold and two-fold symmetry axes, consistent with a cubic arrangement of the chimallin protomers in the particles. Thus, we subjected the particle images to 3D refinement using the subtomogram average obtained from above as an initial reference lowpass filtered to 35 Å and $O$ point-group symmetry enforced, which resulted in a reconstruction at an estimated resolution of 4.2 Å. However, the reconstruction did not display features consistent with this resolution estimate (for example, β-strands were not separated). The high apparent point-group symmetry and distribution of 2D class averages did not support the inflated resolution being due to a preferred orientation. Partitioning particles into half-sets by micrograph did not change the estimated resolution of reconstruction, indicating the inflated estimate was not due to splitting identical or adjacent particles across the half-sets. In addition, extensive 3D classification with and without symmetry enforced did not yield distinct classes. Therefore, the possibility of quasi-symmetry was investigated by performing localized reconstructions of sub-structures within the particles. To reduce computational burden, the apparent $O$ symmetry was first partially expanded to $C4$ using relion_particle_symmetry_expand to fully expand to $C1$ before removing redundant image replicates (noting that redundant views of the four-fold axes possess the same last two Euler angles) to yield 772,788 sub-particles. Refinement of the partially expanded particles while enforcing $C4$ point-group symmetry and using a soft shape mask resulted in a reconstruction with an estimated resolution of 3.6 Å with notably improved features. Re-centreing and re-extraction using a 245 Å side length followed by refinement improved the estimated resolution to 3.6 Å. CTF refinement[44] (per particle defocus, per micrograph astigmatism, beam tilt, and trefoil) and Bayesian polishing[45] successively improved the resolution further to 3.5 Å and 3.4 Å, respectively. A round of 2D-classification without alignment was performed to remove particles assigned to empty or poorly resolved classes, which yielded a set of 664,363 sub-particles and no change in the estimated resolution upon re-running 3D refinement. Although the reconstruction substantially improved through this procedure, the $C4$ map still exhibited distorted density (for example, elongated helices). Attempts at 3D classification did not separate distinct classes. Thus, a localized reconstruction was performed focused on the individual chimallin protomer in $C1$. Again, to reduce computational burden, before expanding the symmetry to $C1$ another round of Bayesian polishing was performed in which the sub-particle images were extracted using a 354 Å side length and premultiplied by their CTF before cropping in real space to a 200 Å side length. After another round of 3D refinement enforcing $C4$ point-group symmetry, the data was expanded to $C1$ which resulted in 2,657,452 sub-particles and refined to an estimated resolution of 3.3 Å. The Bayesian polishing job was re-run to extract sub-particles at the full box size and without premultiplication by their CTF for import into cryoSPARC-v3.2[46]. A single round of local non-uniform refinement[47] was performed in $C1$ using a user-supplied static mask, marginalization, and FSC noise substitution options, which lead to a final reconstruction of the 201phi2-1 chimallin monomer at an estimated resolution of 3.1 Å.

The Goslar chimallin single-particle data were pre-processed similarly to the 201phi2-1 chimallin data, which after initial thresholding

resulted in 2,889 micrographs for further processing. Initial particle positions were identified using the 201phi2-1 chimallin-trained BoxNet2 (Warp-v1.09)[27] model with a threshold of 0.1, which resulted in 289,387 picks. Particles were extracted using a 400 Å side length and subjected to iterative rounds of 2D-classification and sub-selection, which resulted in 78,532 particles used for initial 3D refinement. The Goslar chimallin particles exhibited the same quasi-symmetry as the 201phi2-1 chimallin described above, thus were processed using the same localized reconstruction procedure. The quasi-$O$, quasi-$C4$ and $C1$ reconstructions yielded estimated resolutions of 4.0 Å, 2.6 Å and 2.4 Å, respectively. The quasi-$C4$ and $C1$ reconstructions within RELION[32] were performed on particle images that were extracted using a 470 Å side length, premultiplied by their CTF, and cropped in real space to a 200 Å side length. The final $C1$ reconstruction was performed in cryoSPARC-v3.2[46] as described above, which led to a final reconstruction of the Goslar chimallin monomer at an estimated resolution of 2.3 Å from 1,407,340 sub-particle images.

All resolution estimates are based on the 0.143-cutoff criterion of the Fourier shell correlations between masked independently refined half-maps using high-resolution noise substitution to mitigate masking artefacts[29]. Local resolution estimates were computed using RELION with default parameters. Resolution anisotropy for the $C1$ reconstructions were assessed using the 3DFSC[48] web server which reported sphericity values of 0.963 and 0.994 for the 201phi2-1 and Goslar maps, respectively.

Void peak tilt series were processed similarly to the 201phi2-1 24mer tilt series, but using the gold-fiducials for alignment instead of patch tracking in Etomo[28].

### Coordinate model building and refinement

Initial monomer models were generated via the DeepTracer web server[49] followed by manual building in COOT-v0.9.1[50] and subjected to real-space refinement in PHENIX-v1.19.2[51]. To generate tetramer models, monomer models were rigid-body docked into the $C4$ maps using UCSF Chimera-v1.15[35] and the N-terminal segments joined to the appropriate protomer cores. To generate 24mer models, tetramer models were rigid-body docked into the hexahedral maps and the C-terminal segments reassigned to the appropriate protomer cores. To ensure robust refinement, tetramer and 24mer structures were refined with $C4$ or $O$ non-crystallographic symmetry constraints and reference-model restraints based on high-resolution monomer structures. Isotropic atomic displacement parameters were refined against the respective unsharpened maps. All models were validated using MolProbity[52] and EMRinger[53] (SI Table 2). EMRinger scores for the 201phi2-1 24mer, tetramer, and monomer models were 0.46, 2.86, and 2.39, respectively. EMRinger scores for the Goslar 24mer, tetramer, and monomer models were 0.92, 3.10, and 3.59, respectively.

### Interface analysis

Interface analysis for the cubic assemblies to identify interacting residues and to calculate buried surface area was performed using the ePISA-v1.52[54] and CaPTURE[55] web servers.

### Nine-tetramer sheet modelling

Nine chimallin tetramers were arranged in a flat sheet (3 × 3) structure by fitting in the consensus subtomogram average. Assuming the unfolding of the cubical assembly to create a flat sheet structure, the interacting C-terminal segments in the corner three-fold axis were reassigned in a four-fold symmetry axis to the corresponding protomer. The missing residues between the C-terminal domain and C-terminal segment were built manually in COOT-v0.9.1 ensuring no clash with other modelled atoms (taking the flat sheet model in consideration)[50]. The missing loop region in a protomer (residues 307–319) was built using the DaReUS-Loop web server[56]. This modelled chain (residues 45–612) was used to re-create the flat sheet structure by applying symmetry.

Finally in this flat sheet model, the protruding C-terminal segments of peripheral protomers were trimmed and twelve interacting segments in the periphery were included in the final model (48 chains).

## Protonation state assignment and electrostatics estimates

The electrostatic surface representation was generated with the APBS-v3.0.0[57] using the AMBER99 force field[58] and a pH of 7.5 for assigning protonation states using PROPKA-v3.4.0[59] through PDB-2PQR-v3.4.0[60,61].

## Elastic network models

Elastic network models[61–63] are a subset of normal mode analysis[64,65]. Here we used anisotropic network models (ANM)[66] and Gaussian network models (GNM)[67–69]. Both of these models simplify the protein structure into a series of nodes, with an internode potential energy function governing node motion. To look at it another way, each mode is an eigenvector whose corresponding eigenvalue is the frequency of that motion in the model; lower frequencies correspond to dynamics that best describe the structure's intrinsic motions. ProDy (version 1.0) is a software program enabling calculation of ANM and GNM modes[70,71], which we used in this study. We created 20,412 nodes for the ANM and GNM calculations, which is the largest number of nodes ever used in ProDy.

The five lowest frequency GNM modes accounted for 76% of the overall variance. Considering we do not need to use all ENM modes to capture the system's dynamics[72] we selected these five GNM modes and the five lowest frequency ANM modes to use in our models. The GNM's Kirchoff matrix was built with a pairwise interaction cutoff distance of 10 Å and a spring constant of 1.0, while the ANM's Hessian matrix used a pairwise interaction cutoff distance of 15 Å and a spring constant of 1.0. The ANM structural ensemble movies used an r.m.s.d. difference of 25 Å from the original conformation to display the protein sheet's flexibility.

## Molecular dynamics simulations

Simulations were performed using the nine-tetramer chimallin sheet model. This structure was protonated and placed in a water box through Amber's tleap module[73]. The system was neutralized with $Na^+$ using a 12-6 ion model[74,75]. The CUDA version 10.1 implementation[76–78] of Amber 20 was used[73]. The water model used was OPC[79] with the Amber 19ffsb force field[80]. The resulting system, including the protein and water box, contained 1,729,704 atoms. Energy minimization was performed for a total of 10,000 cycles using a combination of steepest descent and conjugate gradient methods[76] while the heavy atoms were restrained with a force constant of 10.0 kcal mol$^{-1}$ Å$^{-2}$. Next, the system was slowly heated from 10.0 K to 300.0 K over 4 ns before stabilizing at 300.0 K for the next 6 ns using the NVT ensemble with a Langevin thermostat with a friction coefficient (collision frequency) of $\gamma = 5.0$ ps$^{-1}$ and the heavy atoms restrained with a force constant of 1.0 kcal mol$^{-1}$ Å$^{-2}$. Equilibration was performed in the NPT ensemble for 20 ns, using a timestep of 2 fs and the SHAKE algorithm, constraining bonds involving hydrogens[81]. The equilibration temperature was set at 300.0 K with a Langevin thermostat with a friction coefficient[82,83] (collision frequency) of $\gamma = 1.0$ ps$^{-1}$ and the pressure set to 1 bar with a Berendsen barostat[84] with relaxation time constant $\tau = 1.0$ ps$^{-1}$ and a heavy atom restraint with a force constant of 0.1 kcal mol$^{-1}$ Å$^{-2}$. Periodic boundary conditions were enforced with the van der Waals interaction cutoff at 8 Å, while long-range interactions were treated with the Particle mesh Ewald algorithm[85]. After equilibration, the system was cloned into five replicates for the production runs, still set at 300.0 K in the NPT ensemble. Each was run for 300 ns, resulting in 1.5 μs of total sampling.

The resulting molecular dynamics trajectories were analysed through CPPTRAJ-v.25.6[86] and MDTraj-v1.9.4[87]. In particular, r.m.s.d. was calculated with MDTraj. This was done by calculating the r.m.s.d. of the Cα atoms for all tetramers, as well just the central tetramer, from each trajectory and averaging the results (Supplementary Fig. 3).

## Pore analysis

Pore annotation was performed using CHAP-v0.9.1[88] was used for all other analyses. The free energy and solvent density plots were averaged between physiologically identical pores across all simulation replicates. The inter-tetramer (corner four-fold) pore in the upper-left quadrant contained two frames that caused CHAP to crash; these frames were removed before averaging after consultation with the CHAP developers. Considering we still averaged 1,502 frames × 4 pores − 2 bad frames = 6,006 frames for the inter-tetramer pores, we do not feel that this removal causes any difference in our conclusions.

## Structure visualization and figure generation

Density maps, coordinate models and simulation trajectories were visualized and figures were generated with PyMOL-v2.5 (Schrödinger 2021), UCSF Chimera-v1.15[35], ChimeraX-v1.2.5[89], and VMD-1.9.4a35[90].

## Reporting summary

Further information on research design is available in the Nature Research Reporting Summary linked to this paper.

## Data availability

Cryo-EM density maps have been deposited in the Electron Microscopy Data Bank. Subtomogram averaging maps have accession numbers EMD-25221 (201phi2-1, consensus), EMD-25220 (201phi2-1, concave), EMD-25222 (201phi2-1, flat), EMD-25223 (201phi2-1, convex), EMD-25183 (*P. chlororaphis*, 70S), EMD-25229 (Goslar, consensus), EMD-25262 (Goslar, concave), EMD-25358 (Goslar, convex), EMD-25359 (APEC 2248, 70S), EMD-25360 (APEC 2248, 50S). Single-particle maps have accession numbers EMD-25393 (201phi2-1, *O*), EMD-25391 (201phi2-1, *C*4), EMD-25392 (201phi2-1, *C*1), EMD-25393 (201phi2-1, *D*4) EMD-25394 (Goslar, *O*), EMD-25395 (Goslar, *C*4), and EMD-25395 (Goslar, *C*1). Coordinate models have been deposited in the RCSB Protein Data Bank with accession numbers 7SQQ (201phi2-1, *O*), 7SQR (201phi2-1, *C*4), 7SQS (201phi2-1 *C*1 monomer), 7SQT (Goslar, *O*), 7SQU (Goslar, *C*4) and 7SQV (Goslar, *C*1). Raw cryo-EM data have been deposited with the Electron Microscopy Public Image Archive with accession codes EMPIAR-10859 (in situ 201phi2-1 tilt series), EMPIAR-10860 (in situ Goslar tilt series), EMPIAR-10862 (in vitro 201phi2-1 frame series) and EMPIAR-10863 (in vitro Goslar frame series). Genbank IDs for protein sequences used in this study are provided in Supplementary Table 7. All other data are available upon request to the corresponding author(s).

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

**Acknowledgements** All electron microscopy data were collected at the UCSD Cryo-Electron Microscopy Facility, which was built and equipped with funds from UCSD and an initial gift from the Agouron Institute. We thank the UCSD Physics Computing for computational support. We thank V. Lam and J. Hutchings for advice on sample preparation and subtomogram averaging, respectively; J. Krieger and I. Bahar for discussions on elastic network models and expanding ProDy to accommodate the size of the chimallin sheet model; J. Whittman at DMSZ for the gift of Goslar lysates; and members of the Pogliano, Villa, Corbett and Amaro laboratories for helpful discussions and feedback. The authors acknowledge funding from the National Institutes of Health grants R01GM129245 (to J.P. and E.V.), R35 GM144121 (to K.D.C.), and R01GM031749 (to J.A.M.), as well as from the National Science Foundation grants CHE060073 (to R.E.A.) and DBI 1920374 (to E.V.). T.L. is a Simons Foundation Awardee of the Life Sciences Research Foundation. C.S. is supported by a National Science Foundation Graduate Research Fellowship (DGE-1650112). E.V. is a Howard Hughes Medical Institute Investigator. Molecular graphics and analyses were performed in part with UCSF ChimeraX, developed by the Resource for Biocomputing, Visualization, and Informatics at the University of California, San Francisco, with support from National Institutes of Health R01-GM129325 and the Office of Cyber Infrastructure and Computational Biology, National Institute of Allergy and Infectious Diseases.

**Author contributions** Conceptualization: T.G.L., A.D., C.S., R.E.A., J.P., K.D.C. and E.V. Methodology: T.G.L., A.D., J.P., K.D.C. and E.V. Validation: T.G.L., A.D., C.S., R.E.A., J.P., K.D.C. and E.V. Formal analysis: T.G.L., A.D., A.M.P., C.S., R.E.A., J.P., K.D.C. and E.V. Investigation: T.G.L., A.D., A.M.P., C.S., Y.G., E.E., S.S., K.K., E.A.B. and E.A. Data curation: T.G.L., A.D. and C.S. Writing, original draft: T.G.L., A.D., C.S., R.E.A. K.D.C. and E.V. Writing, review and editing: A.M.P., Y.G., E.E., S.S., K.K., E.A.B., E.A., J.A.M. and J.P. Visualization: T.G.L., A.D., A.M.P., C.S., K.D.C. and E.V. Supervision: J.A.M., R.E.A., J.P., K.D.C. and E.V. Funding acquisition: T.G.L., C.S., R.E.A., J.A.M., J.P., K.D.C. and E.V.

**Competing interests** The authors declare no competing interests.

**Additional information**
**Correspondence and requests for materials** should be addressed to Joe Pogliano, Kevin D. Corbett or Elizabeth Villa.

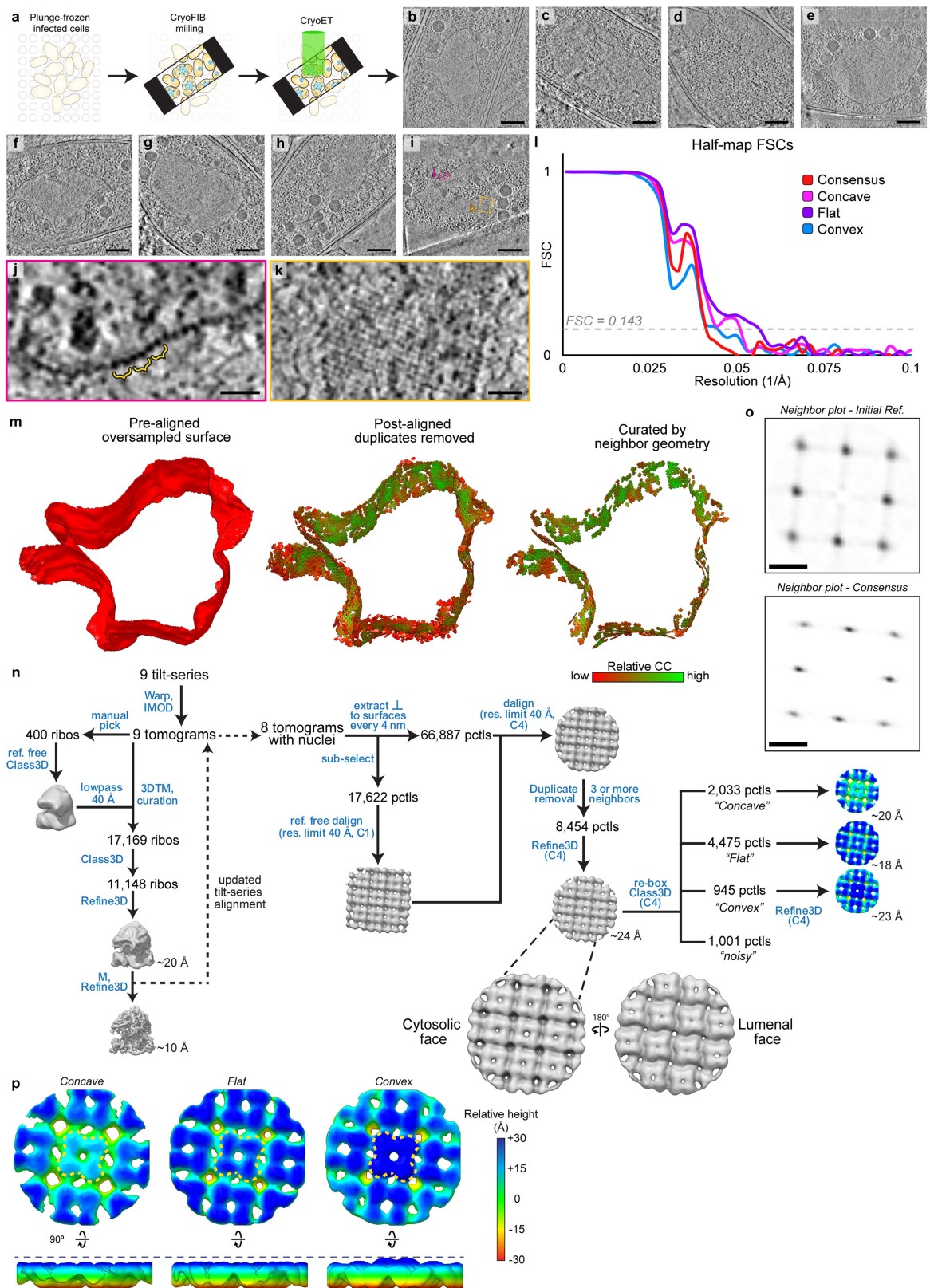

**Extended Data Fig. 1 | *In situ* cryoFIB-ET of 201phi2-1-infected *P. chlororaphis* cells and subtomogram analysis. a**, Schematic of cryoFIB-ET workflow. **b-i**, Slices of the eight 201phi2-1 nucleus-containing tomograms used in this study. **j** Enlarged view of the colored boxed region in **i**. Exemplar doublets are indicated by yellow braces. **k** Enlarged view of the correspondingly colored boxed region in **i** which shows a square mesh-like texture corresponding to the square lattice. **l**, Half-map Fourier shell correlation (FSC) curves for the 201phi2-1 subtomogram reconstructions. **m**, Example over-sampling and subtomogram curation strategy using lattice plots for the tomogram shown in Figure 1c. **n**, Schematic of the subtomogram averaging workflow. Enlarged views of the consensus average with cytosolic and lumeal faces indicated. **o**, Neighbor plot of the initial (top), asymmetrically aligned reference. Neighbor plot of the symmetrized consensus (bottom) refinement. **p**, Enlarged views of the resolved classes colored by relative height. The central tetramer is denoted by a yellow, dashed line for each class. Scale bars: **b–i**: 250 nm, **j,k**: 25 nm, **o**: 10 nm.

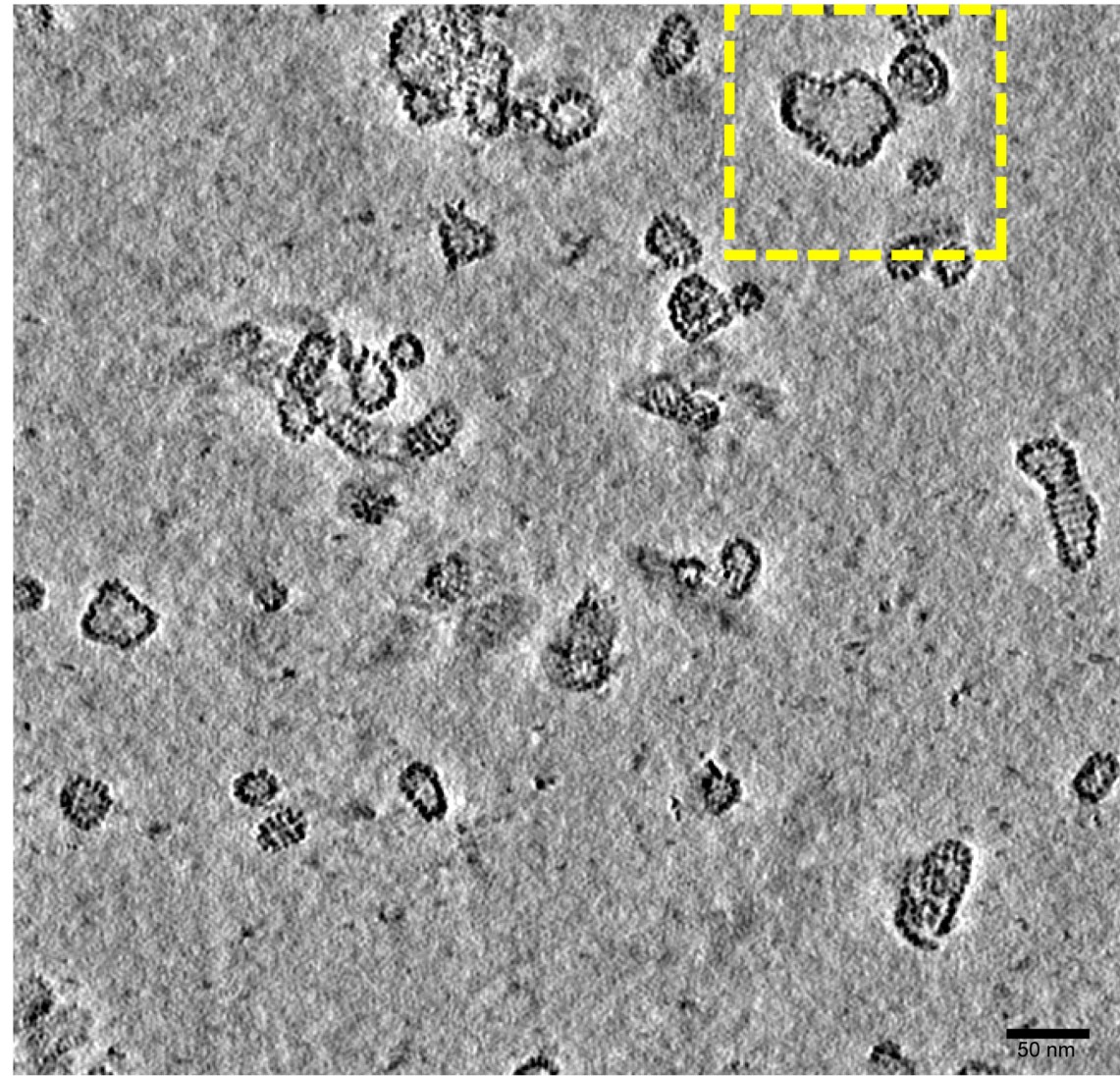

**Extended Data Fig. 2 | In vitro tomography of 201phi2-1 chimallin void peak.** Slice through the tomogram of the 201phi2-1 chimallin SEC size-exclusion chromatography void peak. Region marked by a dashed yellow box is used in Figure 2b. Scale bar is 50 nm.

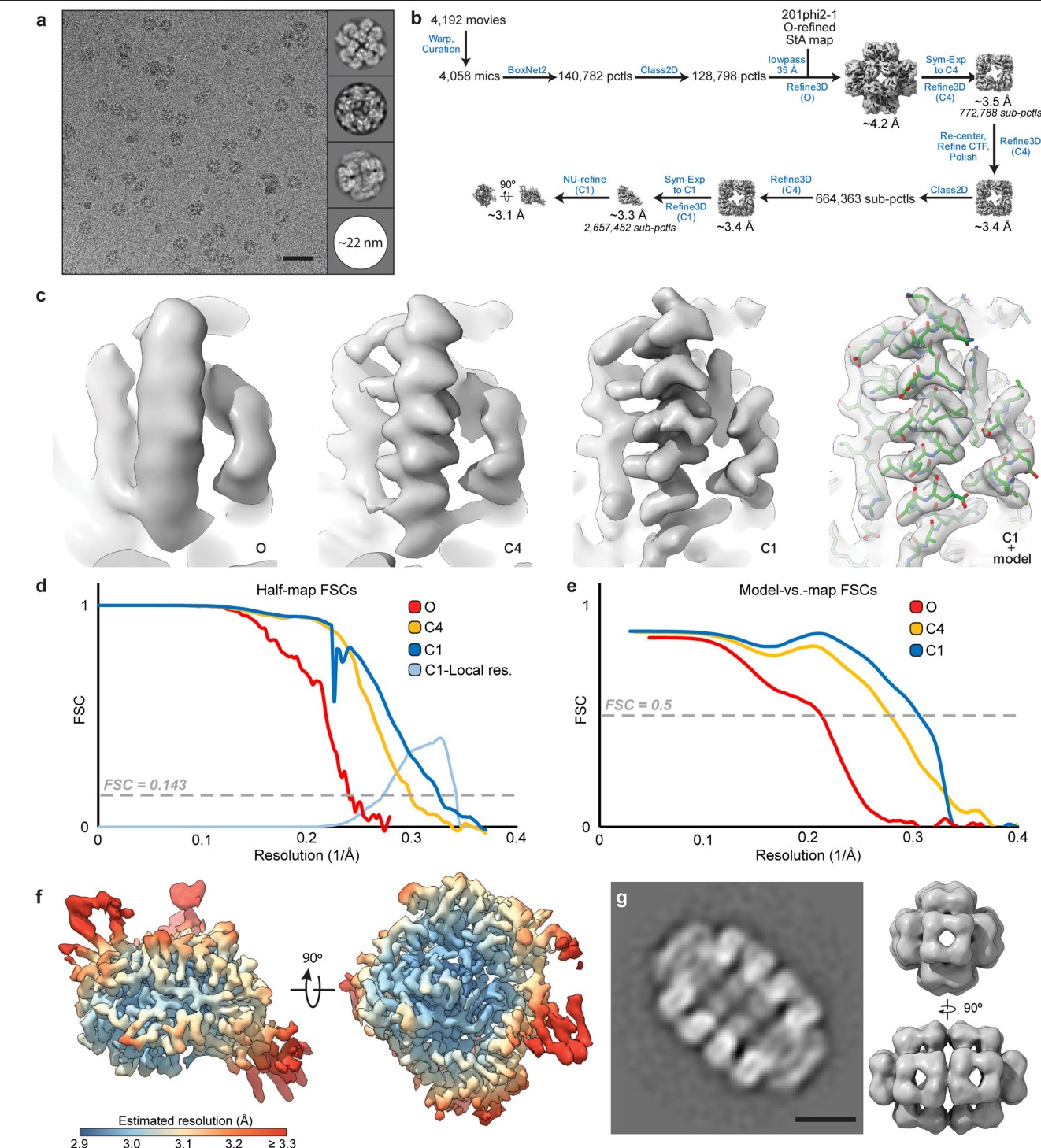

**Extended Data Fig. 3 | Single-particle reconstruction of the *in vitro* 201phi2-1 chimallin cubic assembly. a**, Exemplar micrograph and 2D class averages. **b**, Schematic of the localized reconstruction workflow. **c**, Unsharpened density map views centered on helix B (residues 68-84) at progressive stages of the localized reconstruction process. Final view of the C1 map shown with a fitted coordinate model. **d**, Fourier shell correlation (FSC) curves for the half-maps at progressive stages of the localized reconstruction process (red, yellow, and blue), histogram of local resolution estimates for the C1 reconstruction (light blue). **f**, C1 reconstruction filtered and colored by local resolution. **g**, (left) 2D class average of the minor (517 particles) species of elongated, quasi-D4 assemblies. (right) Orthogonal views of the D4-symmetrized single particle reconstruction. Scale bars: **a**: 50 nm, **g**: 10 nm.

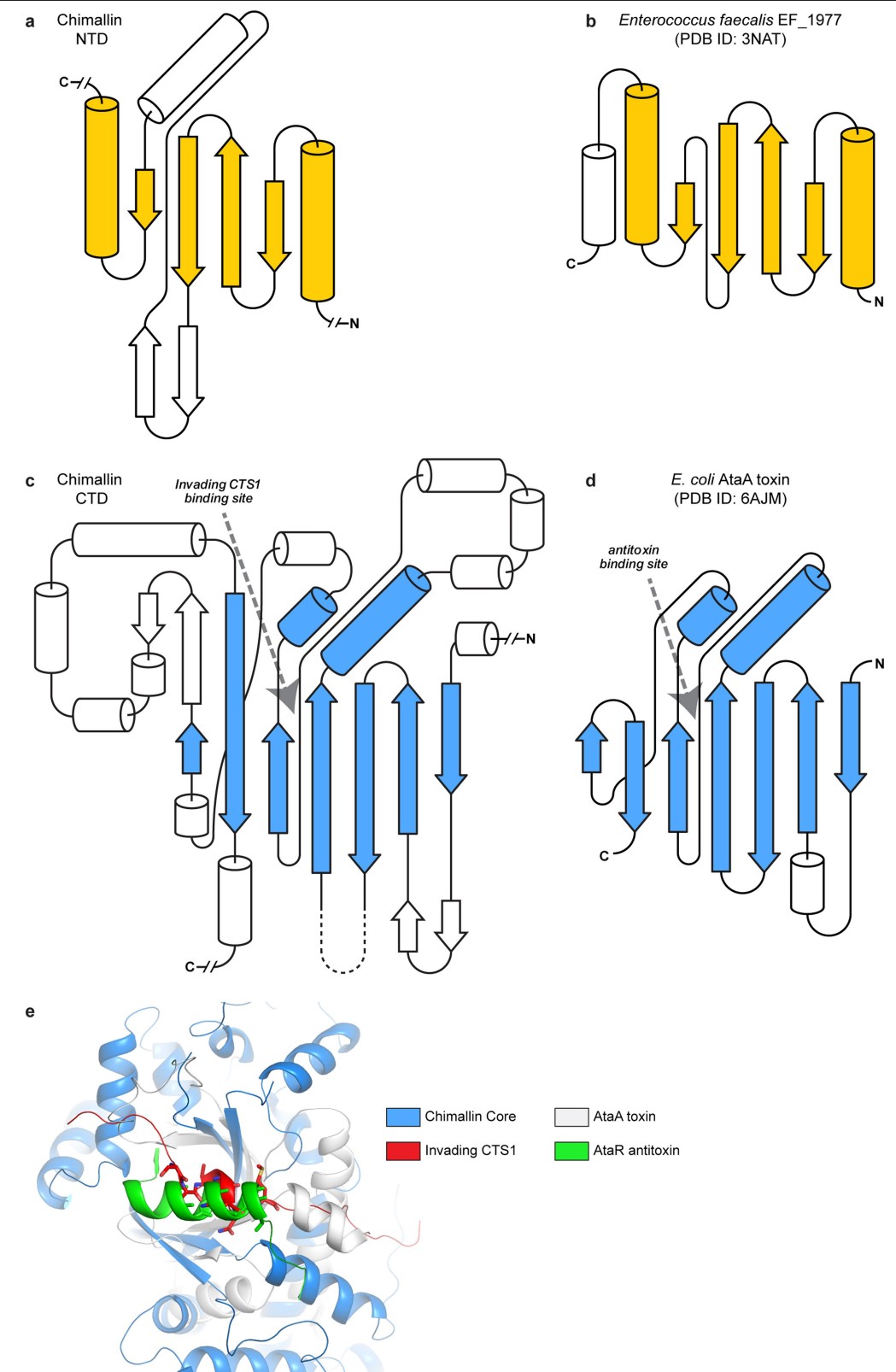

**Extended Data Fig. 4 | Partial homology of chimallin fold topology to known structures. a**, Topology of the 201phi2-1 chimallin N-terminal domain (NTD, residues 62-228). **b**, Topology of *E. faecalis* EF_1977 (PDB ID: 3NAT), the closest structural relative of the chimallin A NTD. The root mean square deviation (RMSD) between chimallin NTD and 3NAT coordinate models is 4.6 Å over 97 aligned Cα atoms. Homologous secondary structure elements are colored in yellow. **c**, Topology of the 201phi2-1 chimallin C-terminal domain (CTD, residues 229-581). **d**, Topology of the *E. coli* AtaT tRNA-acetylating toxin (PDB ID: 6AJM)[11]. The root mean square deviation (RMSD) between chimallin CTD and 6AJM coordinate models is 4.2 Å over 269 aligned Cα atoms. Homologous secondary structure elements are colored in blue. **e**, Structural overlay of the chimallin CTD (blue) and AtaT (white; PDB ID 6AJM), showing the similarity in binding site for the chimallin CTS1 segment (red) and the antitoxin AtaR (green).

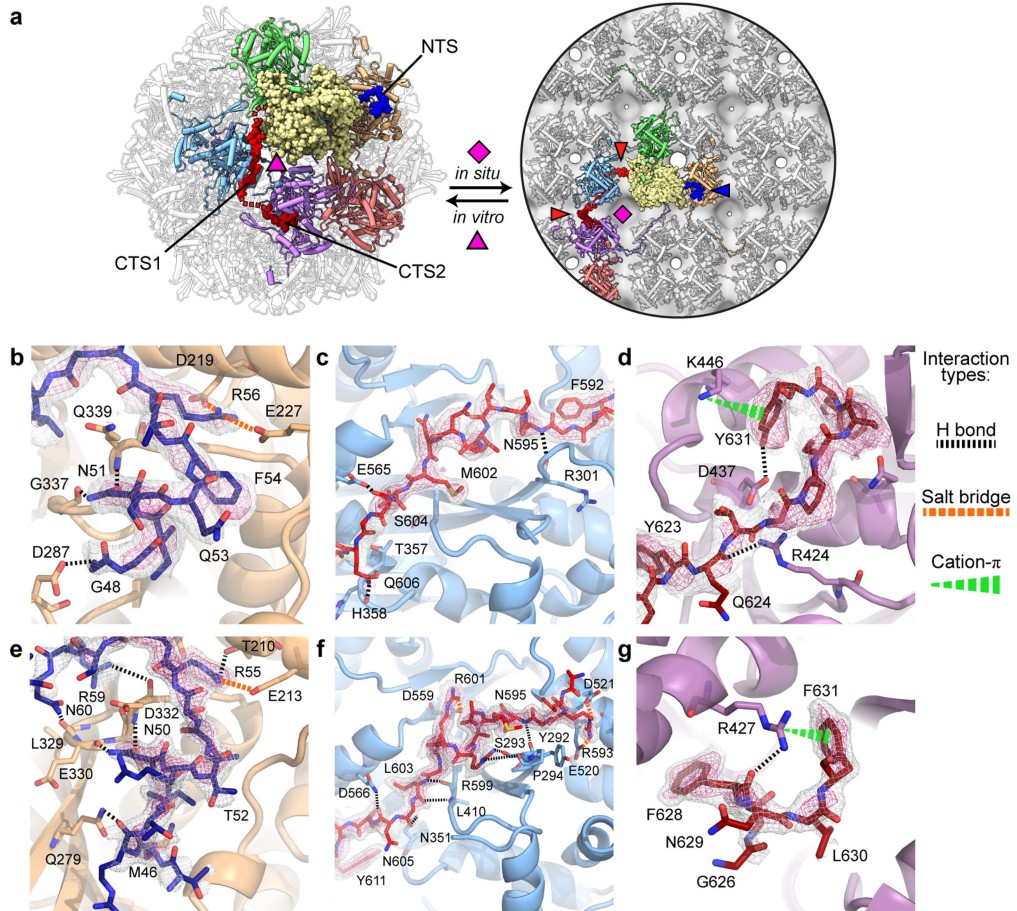

**Extended Data Fig. 5 | Interactions of NTS and CTS segments with the chimallin core. a**, Relationship of 201phi2-1 chimallin protomer packing in the cube (left) and flat sheet model (right). One protomer is shown as spheres and colored yellow with its NTS in blue and CTS1/CTS2 in red. Protomers that interact directly with this central protomer are colored. Non-interfacing protomers are in white. The flat sheet model is docked within the 201phi2-1 consensus subtomogram average map shown as transparent grey. Red arrows point to locations of unresolved linkers (red dashed lines), and pink symbols indicate 3- or 4-fold symmetry axes. b-d, Close-ups of the 201phi2-1 coordinate model around the binding sites for NTS (**b**), CTS1 (**c**), and CTS2 (**d**). (**e**–**g**) Close-ups of the Goslar coordinate model around the binding sites for NTS (**e**), CTS1 (**f**), and CTS2 (**g**). For all panels, cryo-EM density map is shown as a mesh at high (pink) and low (grey) contours. Polar interactions are depicted by the symbols indicated in the key at the far right.

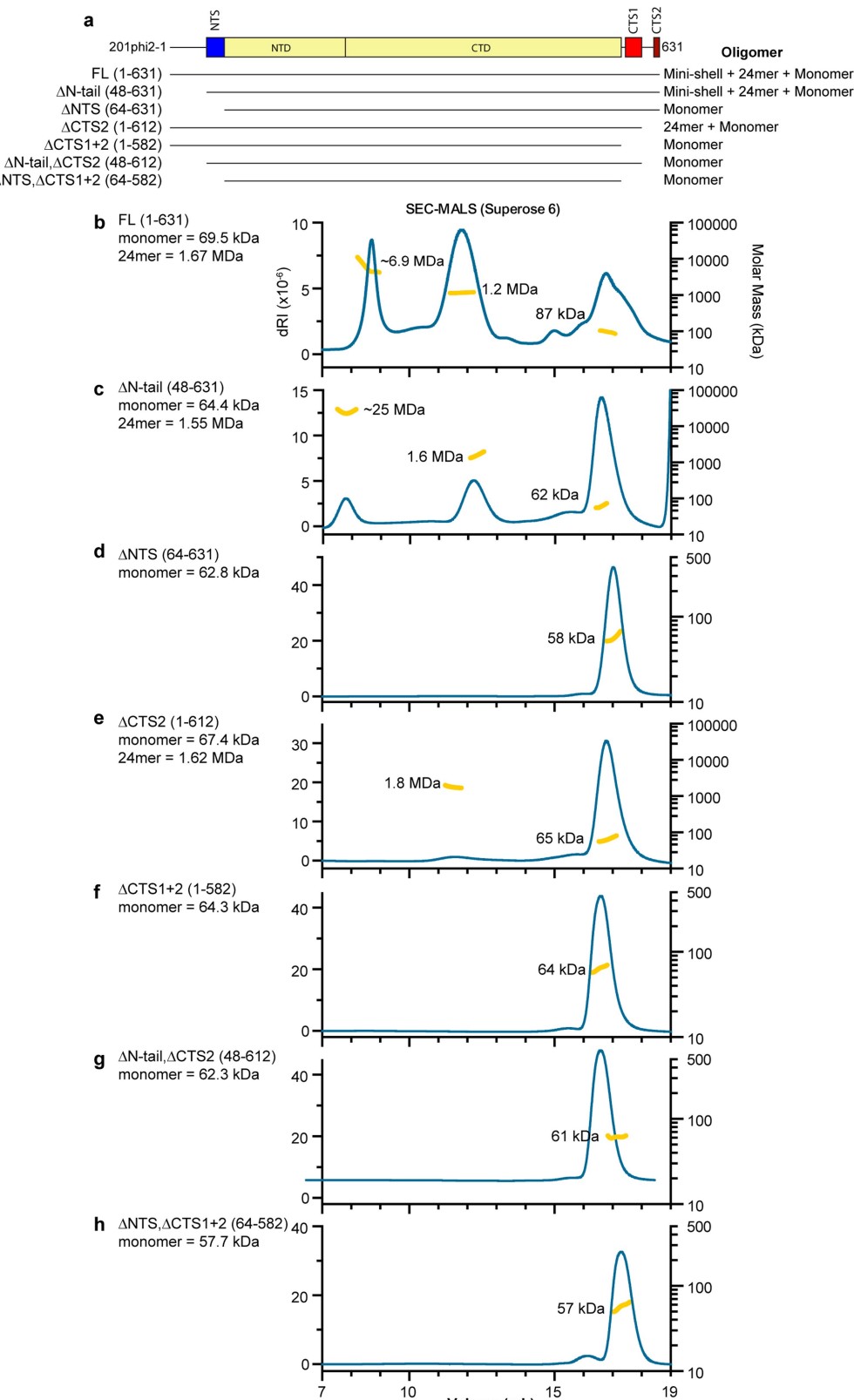

**Extended Data Fig. 6 | SEC-MALS of 201phi2-1 chimallin truncations.**
**a**, Domain diagram of 201phi2-1 chimallin (top), with truncations tested by SEC-MALS (bottom). **b-h**, SEC-MALS analysis of full-length 201phi2-1 chimallin (**b**) and truncated constructs lacking the N-tail (**c**), NTS (**d**), CTS2 (**e**), CTS1+CTS2 (**f**), N-tail+CTS2 (**g**), or NTS + CTS1/2 (**h**). For panels b-h, differential refractive index (dRI) shows protein concentration (blue curves), and yellow points indicate measured molecular weight. Average molecular weight for each peak is shown.

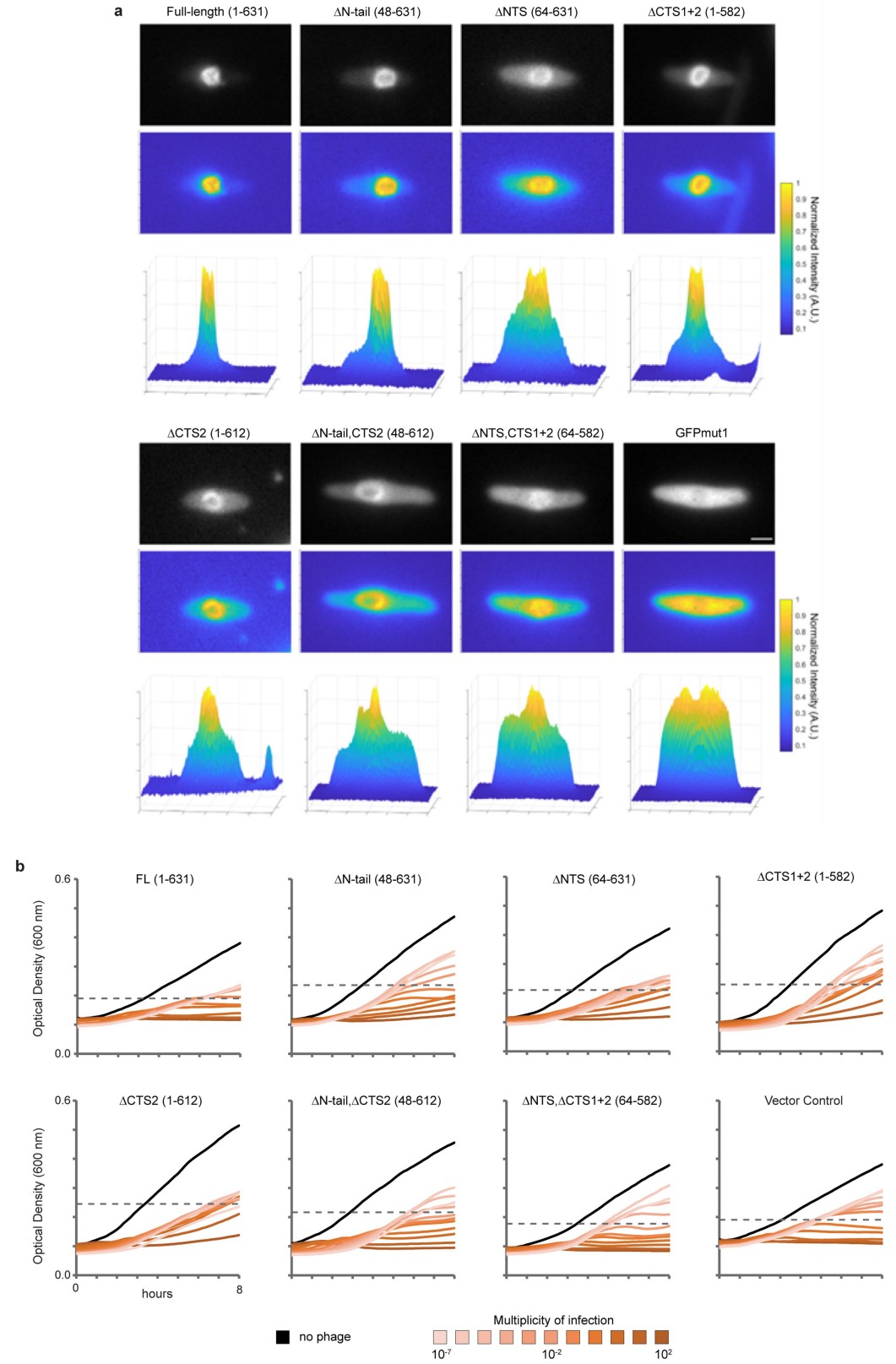

**Extended Data Fig. 7 | GFP-chimallin incorporation into the nuclear shell in 201phi2-1-infected *P. chlororaphis* and truncation mutant growth curves.**
**a**, Raw microscopy images of representative cells expressing GFP-chimallin and infected with 201phi2-1 60 min post-infection (mpi) showing GFP fluorescence with associated 3D graphs showing normalized GFP fluorescence intensity within these cells from a top and side view. GFPmut1 was expressed without fusion to chimallin as a negative control and shows no incorporation. Growth curves for *P. chlororaphis* expressing the indicated 201phi2-1 chimallin truncation mutant (or empty vector control) and challenged with either no phage (black line) or increasing multiplicity of infection of 201phi2-1 (color key at the bottom) over a period of 8 h. Dashed grey-line indicates the half of the maximal optical density at 600 nm achieved by the no phage control in each experiment. Curves are the average of four replicates (n = 4) of each condition. Scale bar: **a**: 1 μm.

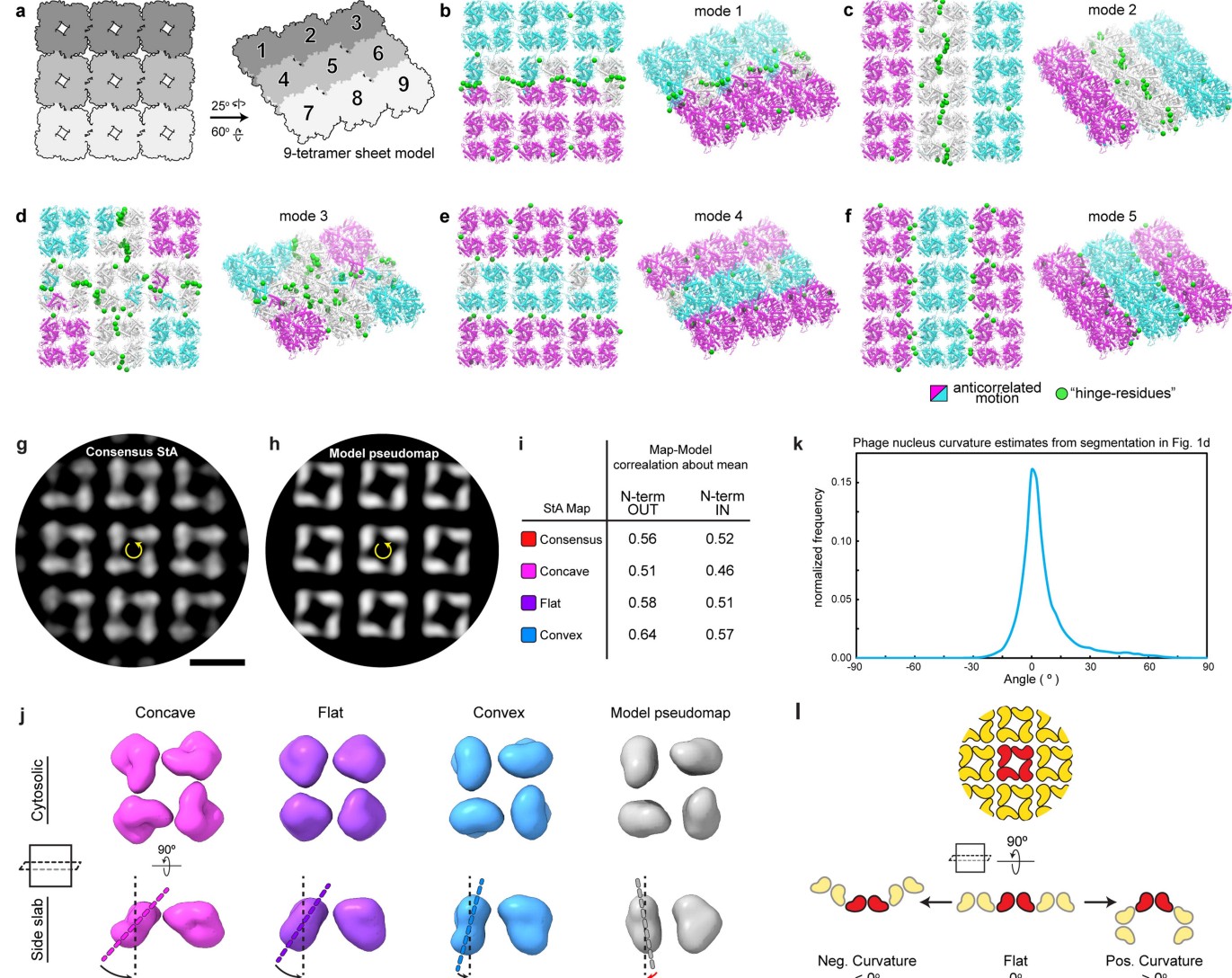

**Extended Data Fig. 8 | Analysis of flexibility by Gaussian network models and subtomogram analysis. a**, Cytosolic and tilted view schematic of 3x3 tetramer sheet model. **b-f**, Cartoon sheet model colored by results of Gaussian Network Model modes 1 through 5, respectively. Regions are colored according to the directional correlation of motion: positive (cyan), negative (magenta), and near-zero (white). "Hinge-residues" are depicted as green spheres. A list of the hinge-residues for each mode is in SI Table 5. **g**, Slice through the consensus subtomogram average for the 201phi2-1 nuclear shell, with the four-fold axis defining the central tetramer noted. **h**, Equivalent view of panel **g** from a pseudomap generating by fitting tetramers into the consensus subtomogram average. **i**, Model-map correlation coefficient about the mean (CC) for a tetramer model fit into the consensus subtomogram average and the three subclasses (concave, flat, and convex) within either the N-termini facing the cytosol (OUT) or lumen (IN). **j**, Two views of the concave, flat, and convex subclasses, compared to a pseudomap generated from the tetramer model. Dotted lines indicate the orientation of one monomer in each map. Denoted angles are with respect to the perpendicular. The arc arrow is red for the pseudomap to denote its opposite direction compared to the subtomogram average maps. **k**, Angles between tetramers in the phage nucleus lattice, derived from surface curvature estimates for phage nucleus segmentation in Figure 1d. **l**, Schematic of example manifestations of lattice curvature. The positive curvature shown on the right represents the 90° angle seen in the *in vitro* cubic assembly.

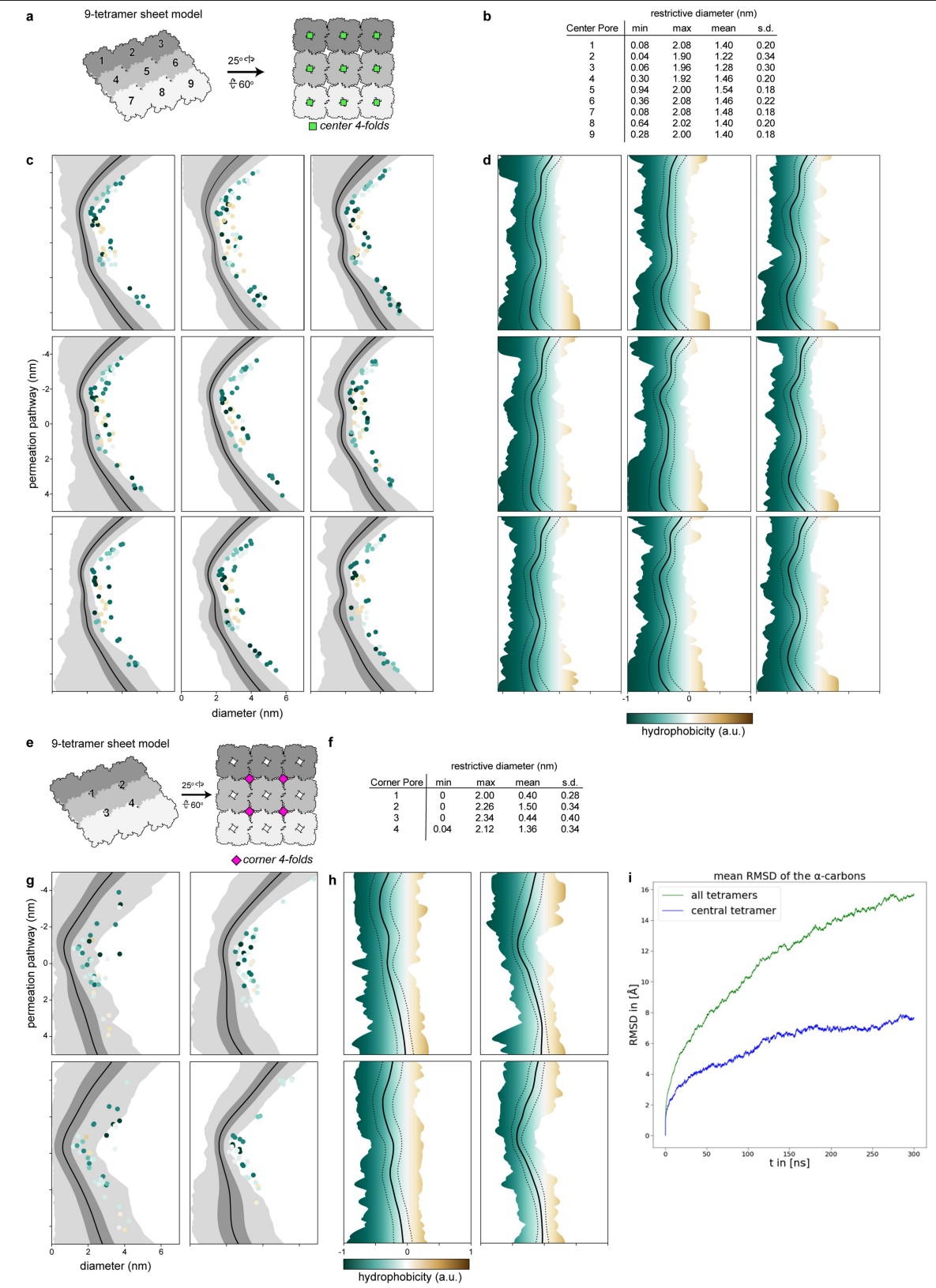

**Extended Data Fig. 9 |** See next page for caption.

**Extended Data Fig. 9 | Size and hydrophobicity profiles of the four-fold pores. a**, Schematic of the 9-tetramer sheet model with the center four-fold pores marked with green squares. **b**, Pore diameter summary statistics for the nine center four-folds denoted in **a** over the course of the averaged 300-ns simulations (n = 5). **c**, Diameter profiles for each pore. The permeation pathway from top (negative values) to bottom (positive values) corresponds with cytosol to lumen. Solid black lines denote the mean diameter, dark gray shading +/− one standard deviation, and light grey shading the range. Dots indicate pore-facing residues and are colored by hydrophobicity. **d**, Hydrophobicity profiles for each pore. Solid black lines denote the mean hydrophobicity, dashed lines +/− one standard deviation, and shaded regions mark the range. **e**, Schematic of the 9-tetramer sheet model with the corner four-fold pores marked with pink squares. **f**, Pore diameter summary statistics for the four corner four-folds denoted in **a** over the course of the averaged 300-ns simulations (n = 5). **g**, Same as **c** for the corner four-fold pores. **h**, Same as **d** for the corner four-fold pores. **i**, Mean root mean square deviation (RMSD) of the alpha-carbons in the 3x3 tetramer sheet model over the course of the simulations for all alpha-carbons (green) and for those just within the central tetramer (blue). The central tetramer is embedded in a physiological environment, flanked by other tetramers. The edge tetramers continue to display an increasing RMSD since they are not connected to adjacent tetramers. Our analysis in the text stems from the central pore and corner pores formed by this tetramer.

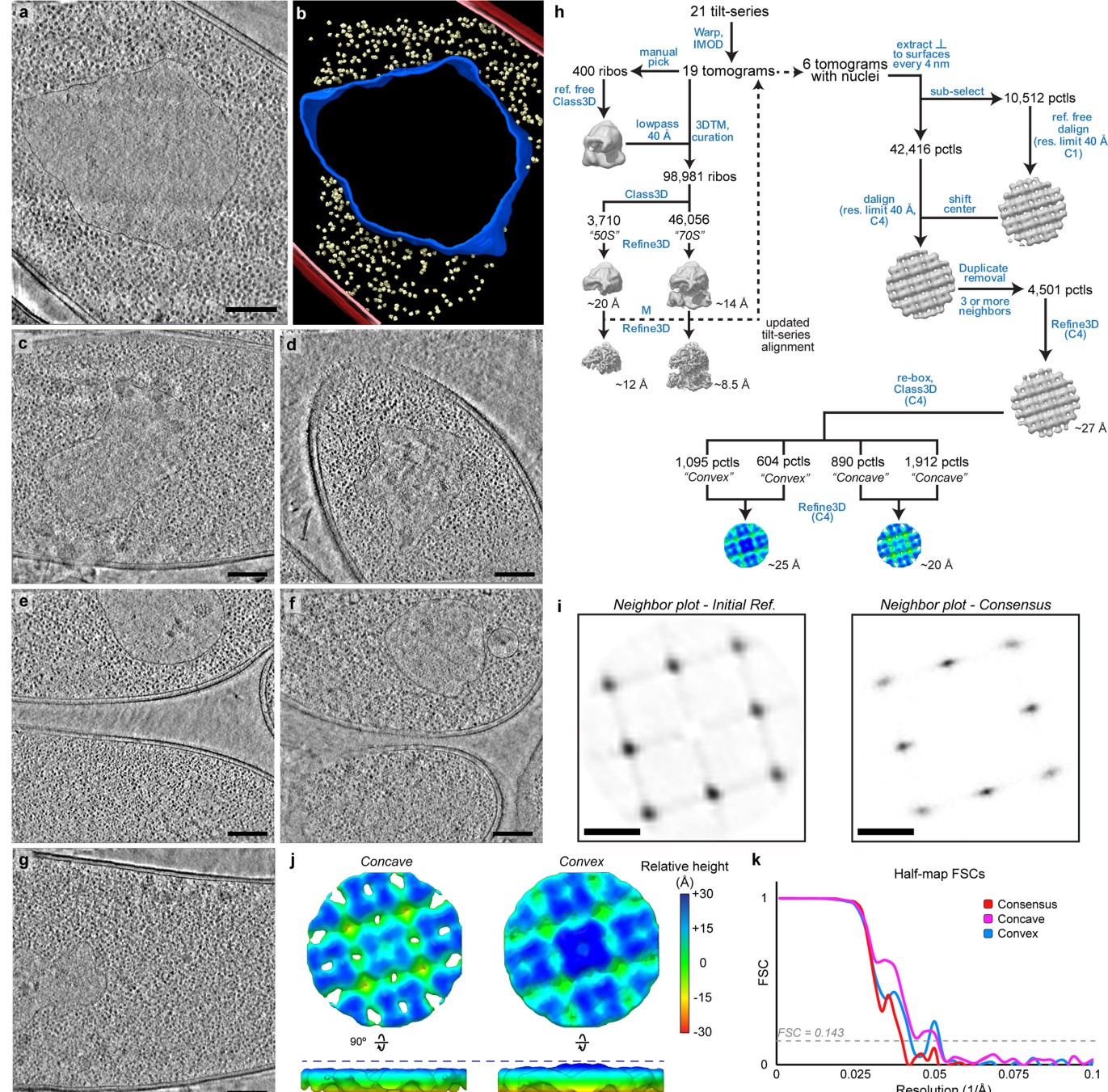

**Extended Data Fig. 10 | *In situ* cryoFIB-ET of Goslar infected APEC2248 cells and subtomogram analysis. a**, Tomographic slice of a Goslar-nucleus and **b** the corresponding segmentation model. Outer and inner bacterial membranes are burgundy and pink, respectively. The phage nucleus is colored blue and host ribosomes are colored pale yellow. Five-hundred randomly selected 70S ribosomes are placed for clarity. **c-g**, A slice from each of the Goslar-nucleus containing tomograms used for subtomogram averaging in this study. The cells were plunged at effectively -20–30 mpi, thus too early to observe virion assembly. **h**, Schematic of the subtomogram averaging workflow. **i**, Neighbor plots of the asymmetrically aligned initial reference (left) and symmetrized consensus refinement (right). **j**, Enlarged views of the resolved classes colored by relative height. **k**, Half-map Fourier shell correlation (FSC) curves for the subtomogram reconstructions.

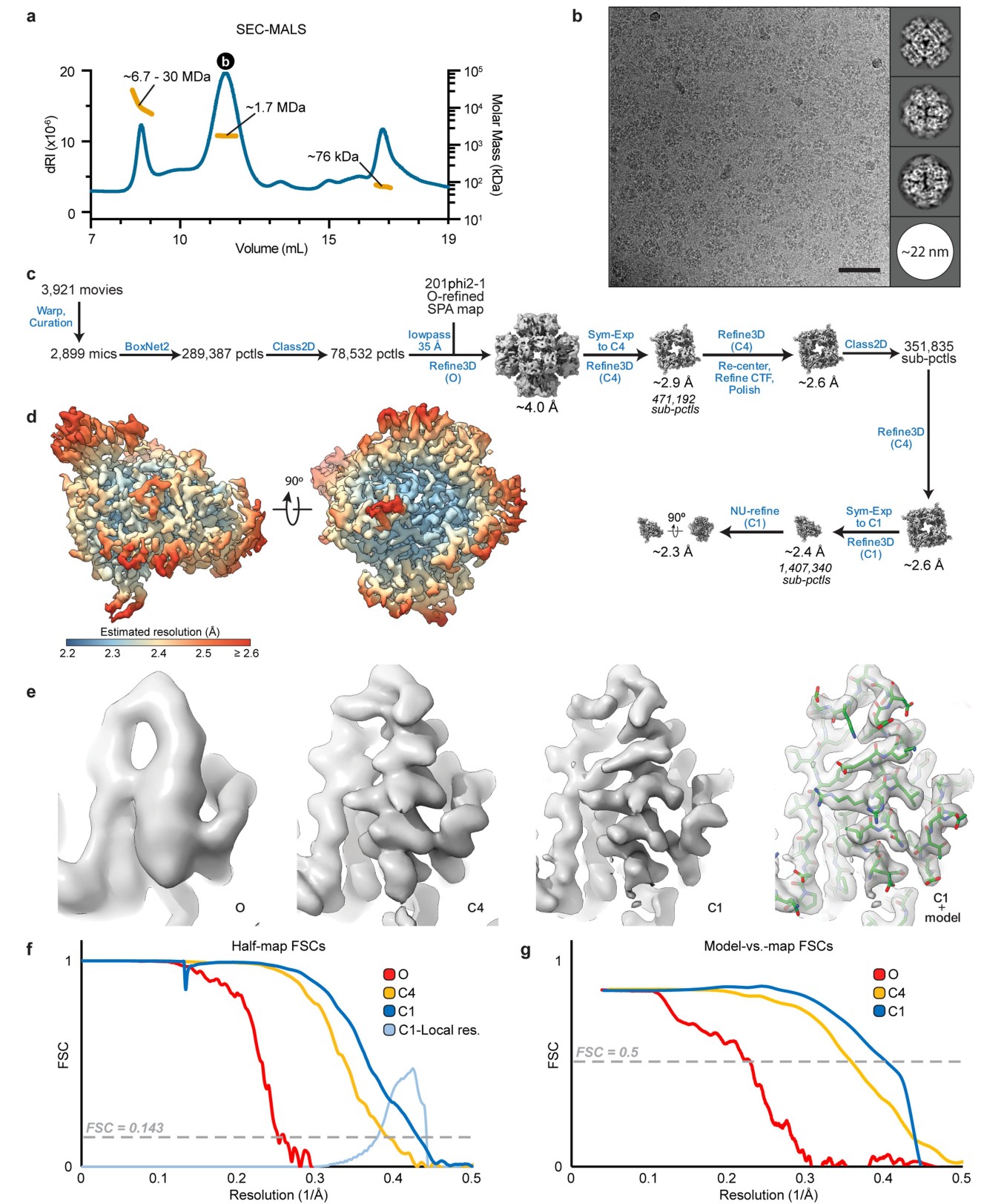

**Extended Data Fig. 11 | Single-particle reconstruction of the *in vitro* Goslar chimallin cubic assembly. a**, Size-exclusion coupled to multi-angle light scattering (SEC-MALS) analysis of purified, full-length Goslar chimallin. **b**, Exemplar micrograph and 2D class averages. **c**, Schematic of the localized reconstruction workflow. **d**, C1 reconstruction filtered and colored by local resolution estimates. **e**, Unsharpened density map views centered on helix B (residues 64-78) at progressive stages of the localized reconstruction process. Final view of the C1 map shown with a fitted coordinate model. **f**,**g**, Fourier shell correlation (FSC) curves for the half-maps and against corresponding models at progressive stages of the localized reconstruction process (red, yellow, and blue), histogram of local resolution estimates for the C1 reconstruction (light blue), and the C1 model-vs-map FSC curve (black). Scale bar: **b**: 50 nm.

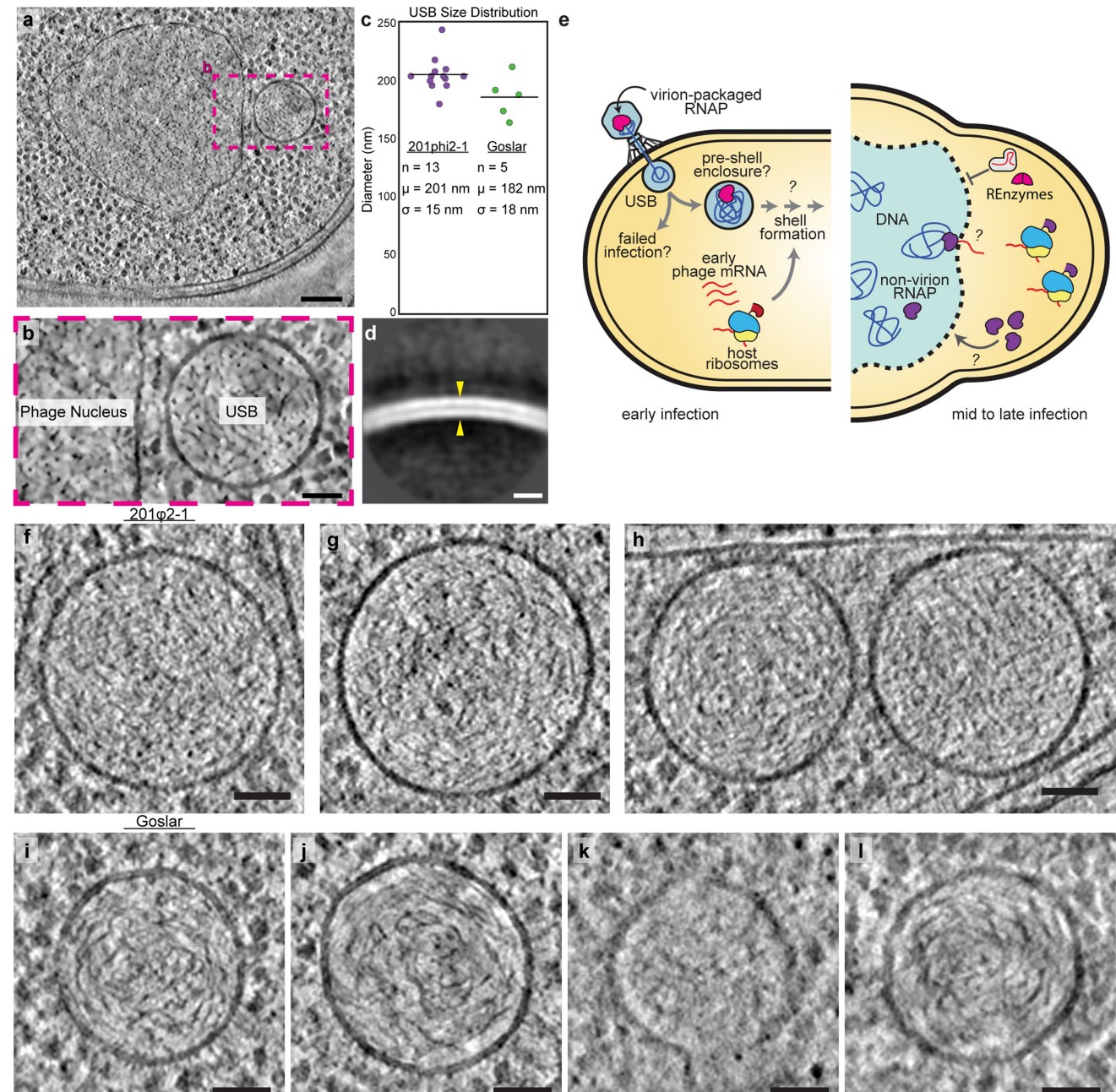

**Extended Data Fig. 12 | Unidentified spherical bodies present in jumbo phage-infected cell populations and speculative models. a**, Tomographic slice of Goslar-infected APEC2248 cell containing a bonafide phage nucleus, as well as an unidentified spherical body (USB). **b**, Enlarged view of the phage nucleus and USB from the region boxed in **a**. **c**, Plot of the apparent maximal diameter distributions for 201phi2-1 (purple) and Goslar (green) USBs with the summary statistics listed. **d**, Subtomogram average of the USBs picked from the Goslar dataset. Yellow arrow pointing to putative membrane leaflets. slices of USBs from the Goslar dataset. **e**, Left, model of USBs as the previously proposed pre-shell/nucleus enclosure of the phage DNA[3]. Right, schematic summary of structural models in this work: (i) exclusion of host nucleases by small chimallin pore sizes, (ii) possible extrusion of phage mRNA via these pores, and (iii) implication of additional shell components to enable uptake of specific phage proteins into the phage nucleus. **f-h**, Gallery of USBs observed in tomograms of 201phi2-1-infected cell populations. **i–l**, Gallery of USBs observed in tomograms of Goslar-infected cell populations. Scale bars: **a**: 150 nm, **b**: 50 nm, d: 10 nm, **f–l**: 50 nm.

Kevin D. Corbett,
Elizabeth Villa

# Reporting Summary

## Statistics

For all statistical analyses, confirm that the following items are present in the figure legend, table legend, main text, or Methods section.

| n/a | Confirmed | |
|---|---|---|
| ☐ | ☒ | The exact sample size ($n$) for each experimental group/condition, given as a discrete number and unit of measurement |
| ☐ | ☒ | A statement on whether measurements were taken from distinct samples or whether the same sample was measured repeatedly |
| ☐ | ☒ | The statistical test(s) used AND whether they are one- or two-sided<br>*Only common tests should be described solely by name; describe more complex techniques in the Methods section.* |
| ☒ | ☐ | A description of all covariates tested |
| ☐ | ☒ | A description of any assumptions or corrections, such as tests of normality and adjustment for multiple comparisons |
| ☐ | ☒ | A full description of the statistical parameters including central tendency (e.g. means) or other basic estimates (e.g. regression coefficient) AND variation (e.g. standard deviation) or associated estimates of uncertainty (e.g. confidence intervals) |
| ☐ | ☒ | For null hypothesis testing, the test statistic (e.g. $F$, $t$, $r$) with confidence intervals, effect sizes, degrees of freedom and $P$ value noted<br>*Give P values as exact values whenever suitable.* |
| ☒ | ☐ | For Bayesian analysis, information on the choice of priors and Markov chain Monte Carlo settings |
| ☒ | ☐ | For hierarchical and complex designs, identification of the appropriate level for tests and full reporting of outcomes |
| ☒ | ☐ | Estimates of effect sizes (e.g. Cohen's $d$, Pearson's $r$), indicating how they were calculated |

*Our web collection on statistics for biologists contains articles on many of the points above.*

## Software and code

Policy information about availability of computer code

| Data collection | DeltaVision softWoRx program-v6.5.2, SerialEM-v3.8b11, ASTRA v.8 |
|---|---|
| Data analysis | DeltaVision softWoRx program-v6.5.2, FIJI-v2.1.0/1.53c, MATLAB-2019a, IMOD-v4.10.28, Warp/M-v1.09, Cube-commit-aa59444, RELION-v3.1.1, Dynamo-v1.1.514,PlaceObjects-1.0.0, UCSF-Chimera-v1.15, dynamo2m-v0.2.2, TomoSegMemTV-vApr2020, Amira-v6.7, PyCurv-commit-fa70ce7, ASTRA v. 8 software, Prism-v9.3, cryoSPARC-v3.2, 3DFSC-webserver-v3, DeepTracer-webserver-v1, COOT-v0.9.1, PHENIX-v1.19.2, MolProbity-v4.5.1, EMRinger-v1.0.0, ePISA-v1.5254 web server, CaPTURE-webserver-v1, DaReUS-Loop-v1, APBS-v3.0.0, PROPKA-v3.4.0, PDB2PQR-v3.4.0, ProDy-v1.0, Amber 19ffsb (Amber 20), CPPTRAJ-v.25.6, MDTraj-v1.9.4, CHAP-v0.9.1, PyMOL-v2.5, ChimeraX-v1.2.5, VMD-1.9.4a35, SHAKE (Amber 19ffsb), Particle mesh Ewald (Amber 19ffsb), Langevin thermostat (Amber 19ffsb), Berendsen barostat (Amber 19ffsb). |

For manuscripts utilizing custom algorithms or software that are central to the research but not yet described in published literature, software must be made available to editors and reviewers. We strongly encourage code deposition in a community repository (e.g. GitHub). See the Nature Portfolio guidelines for submitting code & software for further information.

## Data

Policy information about availability of data

All manuscripts must include a data availability statement. This statement should provide the following information, where applicable:
- Accession codes, unique identifiers, or web links for publicly available datasets
- A description of any restrictions on data availability
- For clinical datasets or third party data, please ensure that the statement adheres to our policy

Cryo-EM density maps have been deposited in the Electron Microscopy Data Bank. Subtomogram averaging maps have the accession numbers: EMD-25221

(201phi2-1, consensus), EMD-25220 (201phi2-1, concave), EMD-25222 (201phi2-1, flat), EMD-25223 (201phi2-1, convex), EMD-25183 (P. chlororaphis, 70S), EMD-25229 (Goslar, consensus), EMD-25262 (Goslar, concave), EMD-25358 (Goslar, convex), EMD-25359 (APEC2248, 70S), EMD-25360 (APEC2248, 50S). Single-particle maps have the accession numbers: EMD-25393 (201phi2-1, O), EMD-25391 (201phi2-1, C4), EMD-25392 (201phi2-1, C1), EMD-25393 (201phi2-1, D4) EMD-25394 (Goslar, O), EMD-25395 (Goslar, C4), and EMD-25395 (Goslar, C1). Coordinate models have been deposited in the RCSB Protein Data Bank with the accession numbers 7SQQ (201phi2-1, O), 7SQR (201phi2-1, C4), 7SQS (201phi2-1 C1 monomer), 7SQT (Goslar, O), 7SQU (Goslar, C4), and 7SQV (Goslar, C1). Raw cryo-EM data have been deposited with the Electron Microscopy Public Image Archive with accession codes: EMPIAR-10859 (in situ 201phi2-1 tilt-series), EMPIAR-10860 (in situ Goslar tilt-series), EMPIAR-10862 (in vitro 201phi2-1 frame-series), and EMPIAR-10863 (in vitro Goslar frame-series). Genbank IDs for protein sequences used in this study are provided in SI Table 7. All other data are available upon request to the corresponding author(s).

# Field-specific reporting

Please select the one below that is the best fit for your research. If you are not sure, read the appropriate sections before making your selection.

☒ Life sciences ☐ Behavioural & social sciences ☐ Ecological, evolutionary & environmental sciences

For a reference copy of the document with all sections, see nature.com/documents/nr-reporting-summary-flat.pdf

# Life sciences study design

All studies must disclose on these points even when the disclosure is negative.

| | |
|---|---|
| Sample size | Sample sizes for were not predetermined and were set by instrument and computational resource availability, as well as the area available for imaging per session in the case of microscopy experiments. |
| Data exclusions | 1) Poorly aligned tilt-series and micrographs were excluded based on CTF-fit quality, as these data are known to negatively impact the quality of subsequent averaging and analysis procedures. 2) The intertetramer ("corner four-fold") pore in the upper-left quadrant contained two frames that caused CHAP to crash; these frames were removed before averaging after consultation with the CHAP developers. Considering we still averaged 1502 frames x 4 pores - 2 bad frames = 6006 frames for the intertetramer pores, we do not feel that this removal causes any difference in our conclusions. |
| Replication | Cryo-FIB/ET of infections was performed independently twice with reproducible results. Cryo-EM samples were technically replicated four times. Fluorescence microscopy and growth curve experiments were performed independently twice for each sample with at least two technical replicates in each instance (specific values indicated in manuscript). SEC-MALS were replicated from independent sample preparations.Molecular dynamics simulations were initialized for five runs and resulted in similar, reproducible trajectories. |
| Randomization | Half-sets for the single-particle cryo-EM data were assigned randomly automatically by RELION-v3.1.1 during the first 3D-auto refinement run. Half-sets for the subtomogram analysis were composed on a per-surface basis in order to achieve roughly even sets. This non-random assignment is necessary as the close-packing of the lattice leads to adjacent subtomograms containing some overlapping or identical information and noise. Thus, splitting adjacent subtomograms randomly into half-sets has a high chance of invalidating assumptions of the method used for resolution estimation, which assumes noise is independent between the half-sets. |
| Blinding | Blinding is not relevant to this study. Researchers were not blinded to the identity of the samples as it was not technically or practically feasible to do so. For cryo-EM experiments at the resolutions achieved in this study, prior knowledge of the sample is essential for reliable interpretation of density features. |

# Reporting for specific materials, systems and methods

We require information from authors about some types of materials, experimental systems and methods used in many studies. Here, indicate whether each material, system or method listed is relevant to your study. If you are not sure if a list item applies to your research, read the appropriate section before selecting a response.

## Materials & experimental systems

| n/a | Involved in the study |
|---|---|
| ☒ ☐ | Antibodies |
| ☒ ☐ | Eukaryotic cell lines |
| ☒ ☐ | Palaeontology and archaeology |
| ☒ ☐ | Animals and other organisms |
| ☒ ☐ | Human research participants |
| ☒ ☐ | Clinical data |
| ☒ ☐ | Dual use research of concern |

## Methods

| n/a | Involved in the study |
|---|---|
| ☒ ☐ | ChIP-seq |
| ☒ ☐ | Flow cytometry |
| ☒ ☐ | MRI-based neuroimaging |

