## [Peer Review File · Nature]

Manuscript Title: Architecture and self-assembly of the jumbo bacteriophage nuclear shell

Reviewer Comments & Author Rebuttals

Reviewer Reports on the Initial Version:

Referees' comments:

Referee #2 (Remarks to the Author):

In this paper, Laughlin and colleagues use state-of-the-art experimental and computational methods to reveal the structure and molecular function of the bacteriophage nuclear shell chimallin protein. The authors fully implement a multiscale approach where electron tomography provides a macroscopic view of the phage nucleus, while cryo-EM is used to dig into the structure of the nuclear shell protein chimallin. The dynamics of the complex are explored through molecular simulations.

This contribution appears to be the first attempt to determine the structure and dynamics of the bacteriophage nuclear shell through a multiscale approach. This contribution thereby represents a stepping stone for the investigation of the mechanics of large bacteriophages. The proposed self-assembly mechanism of the chimallin shell appears unique for the nucleation of this kind of system and could represent a new paradigm for the defense mechanisms of large bacteriophages.

The computational data support the experimental verification of pore formation. If one missing analysis of molecular simulations could be pointed out, this is a detailed analysis of the volume. There are several tools available nowadays. An accurate algorithm designed for pocket volume measure was proposed by Durrant (<https://www.sciencedirect.com/science/article/abs/pii/S1093326310001555>). Detailed analysis of pore volume can provide a valuable quantitative measure to be compared with the experimental data.

A detailed evaluation of the experimental data and procedures goes beyond my expertise. Overall, the paper is very well presented, the experimental data reflect the simulations and the results are solid. The paper is appropriate for publication in Nature upon minor revisions noted above.

Referee #3 (Remarks to the Author):

Here Laughlin, Deep et al. have determined the architecture of the intracellular protein shell produced by jumbo phages during the infection of their hosts. Such nucleus-like compartments protect the replicating viral genomes against several nuclease-based antiviral systems. Among others, they expertly showed the molecular basis of how this viral nuclear shell self-assembles, primarily from a 70kDa protein named chimallin. Using cryo-electron tomography and cryo-electron

microscopy tools, they clearly demonstrated that the assembled structure is a flexible sheet that forms a closed micron-scale compartment. The chimallin lattice has pores, which seems to be large enough to accommodate the transport of some small molecules. The shell structure seems conserved across jumbo phages, although it seems specific to jumbo phages infecting Gram-negative bacteria.

I enjoyed reading the manuscript and the authors provided key novel information on how these unique jumbo phages produce this nucleus-like protein shell to protect their DNA genome during replication. I have made several suggestions which hopefully will improve an already excellent work. I did make some comments regarding some aspects related to phage biology.

1. I strongly suggest replacing the phi symbol by “phi” throughout the manuscript as the ICTV is leaning toward removing all greek letters in phage name (see PMID: 28368359)
2. Lines 24, 282 : Should be CRISPR-Cas
3. Line 24 : bacteriophages should be plural.
4. Line 39: Since you indicated genome size, perhaps also indicate virion size as well.
5. Line 42: Suggestion: “This micron-scale nucleus-like compartment forms ...”.
6. Line 45: It is either capsids or heads but not capsid heads. I suggest capsids.
7. Line 46/48/52: Phage nucleus/nuclear, consider revising the terminology. Perhaps use protein-based shell or shell (as in line 46) or nucleus-like. Check throughout the manuscript.
8. Line 62: Might worth mentioning the time usually needed for this jumbo phage to lyse its host as it would give a meaning to the selected 50-60 minutes time point.
9. Line 63: Figure 1b indicate 45 minutes (and not 50-60 minutes).
10. Line 85: Perhaps indicate the number of aa residues.
11. Line 98: Perhaps indicate the mean here as in line 1047.
12. Line 125: How many residues are we talking about here?
13. Line 127: Any remnant of the gene coding for an anti-toxin AtaR-like in the genome of this phage? Probably not, just asking out of curiosity.
14. Line 174: Have you measured the number of phage particles that are released when these truncated chimallin (Fig. 3d) are expressed in the host? One would assume that it would reduce the number of infective particles? It may give you a clearer effect of these truncated chimallin on the biology of this phage.
15. Line 185: If the luminal face is indeed positively-charged this would probably stabilize the capsid but would this limit the ejection process? The cited reference seems to indicate that this is the case for RNA viruses mostly. Perhaps this needs to be supported by other references or tone down?
16. Line 219: Understanding that the shell is likely flexible, but any suggestion why it needs to be? Transport of molecules?
17. Line 232: What would the size needed for mRNA molecules to pass?
18. Lines 241, 243: phages (plural)
19. Line 242: One would assume that Goslar is a jumbo phage, genome size of? What are the genomic similarities between this phage and 201?
20. Line 244: Number of aa residues in this homolog (and others)?
21. Line 246: time point?
22. Line 272: The phages in Table 9 seems to infect only Gram-negative phages. Are homologs of

chimallin-like found in Bacillus jumbo phages? Perhaps the sequence is too diverse. Could be discussed.

23. Line 285-286: Chimallin does seem to form a shell-like structure in vitro (Figure 2b). This should be discussed.

24. Line 291: How many USBs did you observed in a given infected cell? When you observed the USB, is there DNA also in the shell?

25. Line 306: What was the multiplicity of infection (MOI) used in your phage infection assay? How was it determined? What do you think will happen if two (or more) jumbo phages infect one cell? One would assume that only one shell would be produced but the cell would contain “two” phage genomes. Could these USB reflect such situation/artefact of multiple phage infection within a single cell?

26. Line 308: Does the shell grow at the same time as phage DNA replication or the shell is made and then the viral genome is replicated? One would assume that there is also a limit to the size of the shell?

27. Line 314: What happens to the shell when the cells burst and new virions are released? Is it a stable structure in vivo and in vitro? Presumably the shell still contains DNA when the cell burst? Is it known?

28. Line 319: One would assume that nucleotide building blocks also enters the shell.

29. Line 338: The Materials and Methods section is 16-page long! Some will have to be moved into the supp mat section.

30. Line 348: What was a typical phage titer of a lysate prepare in such way?

31. Line 350: Was phage Goslar amplified in the same way? Same titers?

32. Line 357: How many cells were inoculated?

33. Line 360: As mentioned previously MOI is needed. Why inoculate with such high phage titer? What was the cell concentration after 3-hour incubation?

34. Line 373: Indicate that APEC is an E. coli strain.

35. Line 374: What was the titer of the lysate? MOI ?

36. Line 375: Why the longer incubation period (1.5 hour) compare to 50-60 minutes.

37. Line 1030: Figure 1a: Any idea how many phages are release per infected cell?

38. Line 1037: Figure 1d: How was estimated “the 500 ribosomes”? Is 400 or 500?

39. Line 1072: Should be “c” instead of “C”.

40. Line 1075: Should be “d” instead of “D”.

41. Line 1075: I suggest to indicated in the legend the residues that are still present in these various chimallin constructs.

42. Line 1230. Point missing at the end of the sentence.

43. References: Check titles, many of them have multiple capital letters.

44. References: Bacterial names should also be in italic: Refs 11, 16, 22.

45. References: Check page numbers: Refs 13, 15, 17, 18, 19, 61, 62, 65.

46. References: Name of journals missing o

Referee #4 (Remarks to the Author):

The MS “Architecture and self-assembly of the jumbo bacteriophage nuclear shell” describes

structural characterization of a class of bacteriophage-encoded proteins (termed chimallins) which form a nucleus-like compartment in a bacterial cell that is infected by a “jumbo” phage. The molecular-to-atomic level description of the pseudo-nucleus is supported by mutagenesis and fluorescence microscopy. The determination and characterization of the structure of the highly irregular pseudo-nucleus wall by cryo-FIB tomography and cryoEM is nothing short of impressive.

However, this scientist is struggling to understand the biological and functional novelty of this work. It appears to be a confirmatory in its nature because gp105 of 201phi2-1 has been identified as the primary candidate for being the pseudo-nucleus forming protein in previous studies (Ref. 2). Second, the square mesh of the pseudo-nucleus was first visualized in Ref. 17 (published in 2020). Third, I do not think a massive MD study is required to show that a porous square mesh is impenetrable to large proteins because the image reconstruction, being an averaging technique, already shows the average size of the pores. Fourth, the cooperativity of chimallin assembly has not been studied in this MS, despite the authors’ claim that they showed that “chimallin cooperatively self-assembles as a flexible sheet into closed micron-scale compartments”.

Thus, despite the massive amount of work integrated into this impressive and fairly concise MS, it does not appear to offer a sufficient advance to our understanding of the function of the pseudo-nucleus.

Here are some additional comments that might improve this MS.

A possible confirmation that chimallin can self-assemble into sheets in addition to “balls” would be subtomogram averaging of cryoET of peak “b” of the chromatogram shown in Fig. 2a. A small fragment of the raw tomogram is shown in Fig. 2b, but why was subtomogram averaging not performed on this dataset?

L. 124. The fraction of atoms participating in the superposition has to be shown in addition to the RMSD. One can trim that fraction to a meaninglessly low number and have a very low RMSD (some alignment programs do just that).

L. 177-179. Sorry, I do not find this argument about cooperativity even remotely convincing. Besides, the behavior of the CTS1+2 mutant is difficult to explain. On the one hand, the linkers appear to be required for the incorporation into the nucleus. On the other hand, the CTS1+2 mutant is a monomer that incorporates into the nucleus just like the WT. Please label the panels in Ext Data Fig. 8 with their trivial names (e.g. CTS1+2).

Extended Data Fig. 10 and associated text. What is considered a “bad” CC here, when we are fitting a round shape into a round shape? Do the best CCs (panel c) stand out against the background?

Molecular dynamics

I strongly suggest condensing all MD-related work, including the description of pore properties, to one or two figures. Move all MD-related atomistic panels into Supplementary Information or publish as a standalone study. Consider moving panels d,g,e,h of Fig. 4 to Ext Figs or Suppl Info so that other panels can be enlarged.

I consider all MD work presented here as highly hypothetical because its foundation is the fitting of an essentially round-shaped protein into a 24 Å or so resolution map. I do not think we can extract atomic-level information from this work unless the simulations are run for milliseconds or longer. To this end, please present the equilibration of your system (e.g. as a standard RMSD plot).

L. 322-325. The prohead protease cleavage of chimallin is mentioned at the very end of the MS. However, this information must have been taken into account when the deletion mutants were designed (CTS1+2?). As mentioned above, in the *in vivo* competition experiment, the CTS1+2 mutant behaves abnormally compared to other linker mutants and this needs to be explained or at least commented on.

Finally, it would be interesting to learn if any of the chimallin deletion mutants affect phage morphogenesis. Assuming that chimallin is not integrated into the phage particle, i.e. the infected cell and phage progeny are unchanged except for the presence of chimallin mutants in addition to the WT in the cell, the mutants would be expected to lower the phage titer. Can this be correlated to the fraction of the mutant chimallin incorporated into the nucleus shown in Fig. 3d?

Author Rebuttals to Initial Comments:

We thank the reviewers for recognizing the importance of our work and for their comments and suggestions, which have helped us greatly improve the manuscript. We have addressed all reviewer comments in the point-by-point response below.

Referees' comments:

Referee #2 (Remarks to the Author):

In this paper, Laughlin and colleagues use state-of-the-art experimental and computational methods to reveal the structure and molecular function of the bacteriophage nuclear shell chimallin protein. The authors fully implement a multiscale approach where electron tomography provides a macroscopic view of the phage nucleus, while cryo-EM is used to dig into the structure of the nuclear shell protein chimallin. The dynamics of the complex are explored through molecular simulations.

This contribution appears to be the first attempt to determine the structure and dynamics of the bacteriophage nuclear shell through a multiscale approach. This contribution thereby represents a stepping stone for the investigation of the mechanics of large bacteriophages. The proposed self-assembly mechanism of the chimallin shell appears unique for the nucleation of this kind of system and could represent a new paradigm for the defense mechanisms of large bacteriophages.

The computational data support the experimental verification of pore formation. If one missing analysis of molecular simulations could be pointed out, this is a detailed analysis of the volume. There are several tools available nowadays. An accurate algorithm designed for pocket volume measure was proposed by Durrant (<https://www.sciencedirect.com/science/article/abs/pii/S1093326310001555>). Detailed analysis of pore volume can provide a valuable quantitative measure to be compared with the experimental data.

A detailed evaluation of the experimental data and procedures goes beyond my expertise. Overall, the paper is very well presented, the experimental data reflect the simulations and the results are solid. The paper is appropriate for publication in Nature upon minor revisions noted above.

We thank the reviewer for the positive feedback. We completely agree with the reviewer in the relevance of a detailed analysis of pore volume, and would like to highlight Figure 4e,h and Extended Data Figures 11 and 12 in the initial manuscript version, which provides this analysis for both center (e) and corner (h) pore types. In figure 4e and 4h, we provide the average diameter along the length of the pores, spanning the luminal to cytosolic faces (shown as black line). In addition, the range of pore diameter along the pore length as sampled in the dynamics and the standard deviation, are shown in light and dark gray shading, respectively. This analysis goes beyond what would be provided by a pocket volume measuring tool (such as POVME, mentioned above), allowing readers to understand with one plot how the pores change shape over time. It also allows us to compute the center and corner pore volumes directly by integration. In the case of chimallin we can therefore see that averaged over the 300 ns simulation, the center pores have a volume of 798 ± 81

nm³, whereas the corner pores have a volume of 1429±227 nm³. We have added these values to the caption of the updated Figure 4.

Referee #3 (Remarks to the Author):

Here Laughlin, Deep et al. have determined the architecture of the intracellular protein shell produced by jumbo phages during the infection of their hosts. Such nucleus-like compartments protect the replicating viral genomes against several nuclease-based antiviral systems. Among others, they expertly showed the molecular basis of how this viral nuclear shell self-assembles, primarily from a 70kDa protein named chimallin. Using cryo-electron tomography and cryo-electron microscopy tools, they clearly demonstrated that the assembled structure is a flexible sheet that forms a closed micron-scale compartment. The chimallin lattice has pores, which seems to be large enough to accommodate the transport of some small molecules. The shell structure seems conserved across jumbo phages, although it seems specific to jumbo phages infecting Gram-negative bacteria.

I enjoyed reading the manuscript and the authors provided key novel information on how these unique jumbo phages produce this nucleus-like protein shell to protect their DNA genome during replication. I have made several suggestions which hopefully will improve an already excellent work. I did make some comments regarding some aspects related to phage biology.

We thank the reviewer for highlighting the relevance of our work, and are grateful for the many insightful comments which were interesting to address.

1. I strongly suggest replacing the phi symbol by “phi” throughout the manuscript as the ICTV is leaning toward removing all greek letters in phage name (see PMID: 28368359)

We have made this change throughout the manuscript.

2. Lines 24, 282 : Should be CRISPR-Cas

Done.

3. Line 24 : bacteriophages should be plural.

Done.

4. Line 39: Since you indicated genome size, perhaps also indicate virion size as well.

Done.

5. Line 42: Suggestion: “This micron-scale nucleus-like compartment forms ...”.

Added; thank you.

6. Line 45: It is either capsids or heads but not capsid heads. I suggest capsids.

Done.

7. Line 46/48/52: Phage nucleus/nuclear, consider revising the terminology. Perhaps use protein-based shell or shell (as in line 46) or nucleus-like. Check throughout the manuscript.

We have made minor terminology changes throughout the manuscript to improve consistency both within the manuscript and with prior work in the field. Because of the prevalence of the term “nuclear shell” in prior literature, we have chosen to retain this term.

8. Line 62: Might be worth mentioning the time usually needed for this jumbo phage to lyse its host as it would give a meaning to the selected 50-60 minutes time point.

We have added: “*typical time to lysis is around 90 mpi*”

9. Line 63: Figure 1b indicate 45 minutes (and not 50-60 minutes).

Figure 1b refers to the panels of fluorescence microscopy images taken at around 45 mpi.

For clarification, we have edited the Figure 1c legend referring to the tomogram to:

“Tomographic slice of a phage nucleus in a 201phi2-1-infected *P. chlororaphis* cell at 50-60 mpi.”

10. Line 85: Perhaps indicate the number of aa residues.

Added.

11. Line 98: Perhaps indicate the mean here as in line 1047.

Added.

12. Line 125: How many residues are we talking about here?

We assume the reviewer is referring to the number of alpha-carbon atoms that were overlaid to arrive at the RMSD value (Ln 126 in the revised manuscript); this has been added to the main text and Extended Data Figure 5 legend.

13. Line 127: Any remnant of the gene coding for an anti-toxin AtaR-like in the genome of this phage? Probably not, just asking out of curiosity.

This is an interesting question. There is no such identifiable gene in the genome.

14. Line 174: Have you measured the number of phage particles that are released when these truncated chimallin (Fig. 3d) are expressed in the host? One would assume that it would reduce the number of infective particles? It may give you a clearer effect of these truncated chimallin on the biology of this phage.

This is a great question. We have performed two tests that both indicate that overexpression of ChmA truncations in infected cells does not significantly affect the life cycle of the phage. First, we measured 201phi2-1 phage infectivity (titer) against *P. chlororaphis* cells expressing a representative subset of the analyzed truncations. As can be seen in the figure below, expression of truncated ChmA does not significantly alter the phage titer. As we note in the text, this is likely because there is abundant phage-encoded full-length chimallin also present in these cells that can assemble the shell even in the presence of truncated chimallin protomers that cannot assemble on their own.

We also performed bacterial growth curves in cells overexpressing the ChmA truncations, which more directly addresses the reviewer's question of how well the phage replicates in the presence of overexpressed ChmA truncations. This analysis is now shown in Supplementary Figure 2. While overexpression of ChmA truncations does reduce phage replication efficiency, the effect is not significant compared to our prior analyses of this type with other jumbo phage proteins (see PMID 33436625, specifically Fig. 4e-h). This analysis was also complicated by the lower efficiency of phage 201phi2-1 infection in liquid culture, compared to infection on agar plates.

15. Line 185: If the luminal face is indeed positively-charged this would probably stabilize the capsid but would this limit the ejection process? The cited reference seems to indicate that this is the case for RNA viruses mostly. Perhaps this needs to be supported by other references or tone down?

We thank the referee for pointing out the limitations in scope of the cited reference. Because our work does not deal directly with either DNA packaging or ejection from the capsid, we have removed this sentence entirely.

16. Line 219: Understanding that the shell is likely flexible, but any suggestion why it needs to be? Transport of molecules?

The shape of the nucleus, and its inferred flexibility likely arises from the need to grow while maintaining a seal from nucleases. To accomplish this, in contrast to capsids, there must be several points in the shell where new subunits are added. In membrane compartments, where the composing molecules can laterally diffuse, adding subunits at various locations does not dictate shape, as local rearrangements can smooth the surfaces. Here, new subunits are added and can bend at the tetramer interfaces, potentially contributing to the observed contorted shapes.

Transport of small molecules likely occurs at the pores within and between tetramers. Thus molecular transport likely depends more on the size and shape of pores, and not the overall shape or flexibility of the lattice *per se*. Proteins and larger molecules are likely transported by specialized proteins incorporated into the chimallin lattice. Accommodating these may indeed contribute to the variable shape of the shell.

17. Line 232: What would the size needed for mRNA molecules to pass?

Examples exist of transport of ssRNA through pores as narrow as ~1.2 nm in the contexts of rotavirus capsids (Ding K et al., ref. 14 in the original manuscript) and the *Mycobacterium smegmatis* porin A (MspA) used for nanopore applications (PMID: 22446694) This diameter is indicated in Ln 236 in the revised manuscript.

18. Lines 241, 243: phages (plural)

Added.

19. Line 242: One would assume that Goslar is a jumbo phage, genome size of? What are the genomic similarities between this phage and 201?

All known nucleus-forming jumbo phages have genomes ~250-350 kb in size; Goslar is among the smallest of these with a genome of 237 kb. Compared to 201phi2-1, Goslar shares a core set of genes that includes several blocks of homology that each encompass several genes. While overall sequence identity between the genomes is difficult to meaningfully quantify, their shared genomic architecture and set of related genes (including chimallin, the tubulin homolog PhuZ, and most identified phage structural proteins) strongly indicates that they are highly related. We have now explicitly indicated that Goslar is a jumbo bacteriophage on line 242.

20. Line 244: Number of aa residues in this homolog (and others)?

Added the length of Goslar gp189 (631 amino acids). All of the homologs are roughly consistent in overall length.

21. Line 246: time point?

We have added “mid-infection” in Ln 247 of the main text. In the methods, we have explained that temperature changes involved in transporting samples to the plunge-freezing apparatus make a precise time of infection difficult to assign for this sample.

22. Line 272: The phages in Table 9 seem to infect only Gram-negative phages. Are homologs of chimallin-like found in Bacillus jumbo phages? Perhaps the sequence is too diverse. Could be discussed.

We have only detected Chimallin and PhuZ genes in jumbo phages that infect Gram-negative hosts. Of course, we cannot exclude that jumbo phage families infecting Gram-positive hosts possess highly diverged versions of these genes.

23. Line 285-286: Chimallin does seem to form a shell-like structure in vitro (Figure 2b). This should be discussed.

We have added this point to this sentence.

24. Line 291: How many USBs did you observe in a given infected cell? When you observed the USB, is there DNA also in the shell?

Noting that we ablate a majority of the cell by FIB-milling (and therefore have an incomplete picture of each cell), we have observed up to four USBs within a single infected cell. In cases when a chimallin shell and USB are both present in the same tomograms, both compartments have a similar interior suggestive of densely-packed nucleic acid (Extended Data Figure 16).

Our prior fluorescence microscopy data showed that the nuclear shell is first observable in early-infected cells as a very small structure (nearly diffraction-limited in light microscopy) colocalized with a focus of DAPI staining.

25. Line 306: What was the multiplicity of infection (MOI) used in your phage infection assay? How was it determined? What do you think will happen if two (or more) jumbo phages infect one cell? One would assume that only one shell would be produced but the cell would contain “two” phage genomes. Could these USB reflect such situation/artefact of multiple phage infection within a single cell?

For our analyses, we used very high MOIs (10 or more, measured by phage titers) to ensure that most or all cells that we analyze by cryoET have been infected. We have previously observed rare cases where an infected cell has two phage nuclei, suggesting that each injected phage genome could nucleate shell assembly. In our previous work detailing the discovery of this compartment for 201phi2-1 (Chaikerasitak et al. 2017, PMID: 22446694), about 13% of infected cells (n=264) were observed to contain 2-3 chimallin shells by fluorescence light microscopy. On the other hand, we have also previously shown that infection of a single cell by two different phages (201phi2-1 and phiPA3) can occasionally result in assembly of a single “hybrid” nuclear shell, suggesting that genomes from distinct phage particles can become colocalized into a single nucleus-like compartment (Chaikerasitak et al. 2021, PMID 33436625).

We know from previous studies that the phage genome is protected from DNA-targeting host defenses throughout infection (Mendoza et al. 2020, PMID: 31819262 and Malone et al. 2020, PMID: 31819217). The most vulnerable time for the phage genome is when it first enters the cell, i.e., when only a single copy of the genome is present and infection can be terminated by destroying this copy. Thus, while we don't have information about early infection yet, we assume that the genome has to be protected swiftly. If two independent phage infected the same cell, they would presumably each build a protective layer as swiftly as possible. We believe that the occurrence of multiple USBs per cell and the co-existence of USBs with *bona fide* chimallin nuclear shells is a consequence of multiple phage infecting the same host cell.

26. Line 308: Does the shell grow at the same time as phage DNA replication or the shell is made and then the viral genome is replicated? One would assume that there is also a limit to the size of the shell?

By fluorescence microscopy, we have previously observed that the shell is first observable in early-infected cells as a very small structure (nearly diffraction-limited in light microscopy) colocalized with a focus of DAPI staining. By mid-infection, this shell has grown to roughly the size shown in

Figure 1b. We conclude that the shell grows concomitantly with the replication of phage DNA inside it. We have occasionally observed shells that essentially span the width of the cell.

27. Line 314: What happens to the shell when the cells burst and new virions are released? Is it a stable structure in vivo and in vitro? Presumably the shell still contains DNA when the cell bursts? Is it known?

This question is related to the above. Due to the difficulty of capturing a cell in the act of lysis, we cannot comment on the ultimate fate of the phage nuclear shell once lysis occurs. Subjectively, however, we have observed that the shell looks “deflated” when very late-stage infected cells are observed by cryoET. This suggests that the shell does not shrink/disassemble when the enclosed DNA is loaded into capsids. At the current stage, however, these observations are too preliminary for a firm conclusion.

28. Line 319: One would assume that nucleotide building blocks also enter the shell.

Correct. We make this point in the section “The chimallin lattice can accommodate transport of metabolites but not folded proteins”. In the updated manuscript, we have added specific mention of nucleotides and amino acids to this section.

29. Line 338: The Materials and Methods section is 16-page long! Some will have to be moved into the supp mat section.

Nature presents all Methods as online-only.

30. Line 348: What was a typical phage titer of a lysate prepared in such way?

We typically observe a phage titer of 10^{11} - 10^{12} pfu/mL for 201phi2-1.

31. Line 350: Was phage Goslar amplified in the same way? Same titers?

As noted in the Methods, Goslar lysate was prepared by Johannes Wittmann at the DSMZ, as detailed in our related manuscript more completely describing this phage and its infection cycle (*bioRxiv* DOI: 10.1101/2021.10.25.465362). The Goslar phage titer is 1×10^{10} PFU/mL (PMID: 31109012).

32. Line 357: How many cells were inoculated?

We have added this number to the methods: 5 μ L of $OD_{600}=0.6$.

33. Line 360: As mentioned previously MOI is needed. Why inoculate with such high phage titer? What was the cell concentration after 3-hour incubation?

The MOI of this experiment is measured in the hundreds. We have added further details on cell concentration and volume used for the on-pad infections, and these numbers imply an MOI of at least 1000. We observe that the cell density does increase during the three-hour incubation on pads, but we have not precisely measured this increase. Therefore, we can estimate the MOI of these experiments at ~ 200 -500.

We used very high phage titers to ensure a high infection rate, such that the vast majority of cells are infected when examined by cryoET. This method has been optimized over several years, and while

it's an unusually high number for phage infection experiments, the high MOI should not affect the outcome of our findings that occur at the molecular level.

34. Line 373: Indicate that APEC is an *E. coli* strain.
Done.

35. Line 374: What was the titer of the lysate? MOI ?
As detailed in a prior paper from Johannes Wittmann's lab (who provided the Goslar lysates), the titer is 1×10^{10} PFU/mL (PMID: 31109012).

36. Line 375: Why the longer incubation period (1.5 hour) compared to 50-60 minutes.
The longer incubation time for Goslar was based on the fact that Goslar infections of *E. coli* cells proceed more slowly than 201phi2-1 infections of *P. chlororaphis*.

37. Line 1030: Figure 1a: Any idea how many phages are release per infected cell?
The average burst size of 201phi2-1 is 16 (PMID: 22726436).

38. Line 1037: Figure 1d: How was estimated "the 500 ribosomes"? Is 400 or 500?
The 500 ribosomes is an exact value and the population is a random subset of those retained after alignment and classification of template-matching hits.

39. Line 1072: Should be "c" instead of "C".
Done.

40. Line 1075: Should be "d" instead of "D".
Done.

41. Line 1075: I suggest to indicate in the legend the residues that are still present in these various chimallin constructs.
Added.

42. Line 1230. Point missing at the end of the sentence.
Added.

43. References: Check titles, many of them have multiple capital letters.

44. References: Bacterial names should also be in italic: Refs 11, 16, 22.

45. References: Check page numbers: Refs 13, 15, 17, 18, 19, 61, 62, 65.

46. References: Name of journals missing o

We have carefully checked and updated all references.

Referee #4 (Remarks to the Author):

The MS "Architecture and self-assembly of the jumbo bacteriophage nuclear shell" describes structural characterization of a class of bacteriophage-encoded proteins (termed chimallins) which

form a nucleus-like compartment in a bacterial cell that is infected by a “jumbo” phage. The molecular-to-atomic level description of the pseudo-nucleus is supported by mutagenesis and fluorescence microscopy. The determination and characterization of the structure of the highly irregular pseudo-nucleus wall by cryo-FIB tomography and cryoEM is nothing short of impressive.

However, this scientist is struggling to understand the biological and functional novelty of this work. It appears to be a confirmatory in its nature because gp105 of 201phi2-1 has been identified as the primary candidate for being the pseudo-nucleus forming protein in previous studies (Ref. 2).

The reviewer is correct about this point, which we also make in the manuscript. While prior work had shown that this protein is incorporated into the phage nuclear shell, the current study is the first to directly demonstrate this, and to provide structural evidence that it is the primary component of the nuclear shell.

Second, the square mesh of the pseudo-nucleus was first visualized in Ref. 17 (published in 2020).

The referenced study performed an analysis of *Salmonella* cell lysates infected with the jumbo phage SPN3US, and in this analysis they identified fragments of a sheet-like assembly with square packing and dimensions consistent with our study. The analysis of this sheet structure was very limited, giving no information about the underlying component(s) or its internal structure. Further, the authors recognized their inability to establish the identity of the lattice or its role in phage replication, and cited our prior paper describing the phage nuclear shell (Chaikeratisak et al., 2017, PMID 22446694) as a possible candidate for the lattice they observe. Finally, since that analysis was performed on cell lysates, the observed sheet could not be directly linked to the phage nuclear shell. Dramatically changing conditions, like cell lysis and exposure to an air-water interface, can affect the structure of molecular assemblies. Our multiscale analysis directly identifies chimallin as the major phage nuclear shell component, reveals the molecular basis for its assembly into a square lattice, and directly shows that this lattice forms the phage nuclear shell.

Third, I do not think a massive MD study is required to show that a porous square mesh is impenetrable to large proteins because the image reconstruction, being an averaging technique, already shows the average size of the pores.

The reviewer is correct about this point. We would like to point out, however, that the MD simulations and network analyses revealed more than the average size of the pores and the (rather substantial) range of diameters sampled. Notably, the methods applied indicate the likely modes of flexibility afforded by the distinctive square lattice structure assembled by chimallin. Such information would not be readily deduced by observation of the static structures alone.

Fourth, the cooperativity of chimallin assembly has not been studied in this MS, despite the authors' claim that they showed that “chimallin cooperatively self-assembles as a flexible sheet into closed micron-scale compartments”.

Our study shows how chimallin self-assembles into a sheet structure through multivalent interactions between neighboring protomers. Further, our observation that purified chimallin in vitro

forms only monomers and large, closed, shell-like structures of 24 or more subunits (no dimers, trimers, tetramers, etc.) strongly suggests that its assembly is highly cooperative. However, the reviewer is correct that we have not directly measured the cooperativity of chimallin assembly. Therefore, we have removed the word “cooperative” from the manuscript.

It is very important to note here that none of our conclusions depend on the interpretation that chimallin self-assembly is cooperative. The protein clearly forms closed shells both *in vitro* and in infected cells; the dynamics and energetics of assembly, while important, are not strictly relevant to our current conclusions about the architecture and protective functions of the assembled phage nuclear shell.

Thus, despite the massive amount of work integrated into this impressive and fairly concise MS, it does not appear to offer a sufficient advance to our understanding of the function of the pseudo-nucleus.

Here are some additional comments that might improve this MS.

A possible confirmation that chimallin can self-assemble into sheets in addition to “balls” would be subtomogram averaging of cryoET of peak “b” of the chromatogram shown in Fig. 2a. A small fragment of the raw tomogram is shown in Fig. 2b, but why was subtomogram averaging not performed on this dataset?

We attempted subtomogram averaging of “peak b” structures (reviewer figure 1), but could not obtain a meaningful 3D average(s), for several reasons: First, the size and number of these structures in our tomograms was relatively low, limiting the number of subtomograms that could be extracted and averaged. Second, these structures show high local curvature relative to the much larger shells observed in cells, complicating the analysis. Finally, as our goal with *in vitro* studies was to perform high-resolution structural analysis, we focused on the 24mer “balls” (cubes) rather than this fraction. The visual consistency between “peak b” structures and both the “balls” and shells observed in cells - including the square lattice and subunit spacing - clearly links these structures without subtomogram analysis.

Subtomogram analysis of 201phi2-1 chimallin SEC-MALS in Figure 1b. (a) Slice through a representative tomogram of the chimallin SEC void peak. Region marked by a dashed yellow box was used for Figure 1b. Scale bar is 50 nm. (b). Orthogonal views of an example average obtained when attempting to perform subtomogram analysis of chimallin assemblies present in the peak b.

Panel a of the above figure has been added to the revised manuscript as Supplementary Figure 1 and indicated as the source image of Figure 1b in the accompanying SI Guide.

L. 124. The fraction of atoms participating in the superposition has to be shown in addition to the RMSD. One can trim that fraction to a meaninglessly low number and have a very low RMSD (some alignment programs do just that).

The reviewer is correct. We have updated the RMSD numbers in both the text and the figure legend, to include the number of overlaid alpha-carbon atoms.

L. 177-179. Sorry, I do not find this argument about cooperatively even remotely convincing. Besides, the behavior of the CTS1+2 mutant is difficult to explain. On the one hand, the linkers appear to be required for the incorporation into the nucleus. On the other hand, the CTS1+2 mutant is a monomer that incorporates into the nucleus just like the WT.

As explained above, we have now removed the term “cooperativity” from this section and re-phrased this discussion to clarify our conclusions.

All of the truncation mutants shown in Figure 3, with the exception of the removal of residues 1-47 (which are unresolved in our structure), strongly disrupt chimallin self-assembly *in vitro* and are

predominantly monomeric in purified solution (Figure 3c). The analysis of their incorporation into the phage nuclear shell (Figure 3d) is complicated by the presence of abundant phage-encoded wild-type chimallin in addition to the separately-expressed truncation mutants. The wild-type protein always forms a shell, and all of the truncation mutants are incorporated into this shell to varying degrees. The level of incorporation varies between truncations, and the CTS1+2 truncation's incorporation level, while not statistically significantly different from wild-type, is nonetheless reduced. While we cannot clearly explain why the CTS1+2 truncation appears to show slightly higher incorporation into the chimallin lattice than some other mutants, the overall trend is clear that removal of the NTS and CTS segments reduces incorporation into the lattice.

Why are the truncations all incorporated into the phage nuclear shell, despite lacking the NTS and CTS segments? Simply put, all of these truncations retain the surfaces to which NTS and CTS segments from neighboring subunits bind, so in the presence of abundant wild-type protein they are incorporated into the lattice despite their lack of NTS and/or CTS segments.

The statement on lines 177-181 was meant to convey that, even when the chimallin lattice had been "poisoned" by incorporation of some subunits without NTS/CTS segments, it still robustly assembles. We have rephrased this point to more precisely convey this meaning.

Please label the panels in Ext Data Fig. 8 with their trivial names (e.g. CTS1+2).

Done.

Extended Data Fig. 10 and associated text. What is considered a "bad" CC here, when we are fitting a round shape into a round shape? Do the best CCs (panel c) stand out against the background?

It has come to our attention that it is more appropriate to use the correlation about the mean, rather than the default about zero in UCSF-Chimera which is more appropriate for negative-stain maps. In addition, for comparison, we have updated this figure to include scores for docking of models with reversed sidedness. The sidedness placing the N-terminus facing the cytosol is consistently higher for all classes. Values specified in the main text have been updated to reflect this change. While absolute values changed, the trends are still consistent with the observed confirmations and sidedness of chimallin in the context of assembled shells.

Molecular dynamics

I strongly suggest condensing all MD-related work, including the description of pore properties, to one or two figures. Move all MD-related atomistic panels into Supplementary Information or publish as a standalone study. Consider moving panels d,g,e,h of Fig. 4 to Ext Figs or Suppl Info so that other panels can be enlarged.

I consider all MD work presented here as highly hypothetical because its foundation is the fitting of an essentially round-shaped protein into a 24 Å or so resolution map. I do not think we can extract atomic-level information from this work unless the simulations are run for milliseconds or longer. To this end, please present the equilibration of your system (e.g. as a standard RMSD plot).

We followed the suggestion of the reviewer and consolidated the MD and simulation panels into a single figure. We also agree that it is important to provide structural characterization of the system during the simulations, and now also provide the time-series RMSD plots in SI Fig. 3 for both the central tetramer and all nine tetramers. The RMSD of the central tetramer in levels off, showing that we have appropriately equilibrated the system and can derive useful data from it. The RMSD including the tetramers in the edges increases indicating the system has not reached stability, which is due to the fact that at the edges, these tetramers are in an artificial state, not bound by chimallin tetramers in all four sides. Our analysis is made on the central and side pores formed by the central tetramer, which should not be affected by edge effects, as they represent the same environment as in the physiological sheet. We do not attempt to report data encompassing the central pores of the other tetramers, or of any loops or regions we do not feel are properly equilibrated.

We agree in full on the importance of proper statistical sampling. To this end, we note that we simulated five replicas (copies) of the full system, thus we have achieved nearly *1.5 microseconds* of sampling for this very large system. Furthermore, when considering the individual chimallin tetramers, we have simulated 9 copies of the tetramer within each 300-ns simulation, with five replicas. Thus for the tetramer unit we have actually achieved *22.5 microseconds* of dynamical sampling, which - based on standards of the field - is considered sufficient to quantify behavior down to the level of individual atoms.

It is possible that the structure of the loops may (or may not) differ in their exact conformation, although, as these loops are subjected to tens of microseconds of statistical sampling, the analysis contained herein is at least well-sampled. Our initial model together with the MD simulations therefore provides valuable information about several important system features, including the overall structural stability of the modeled system, as well as pore dynamics, including average pore diameter and volume, well into microsecond timescales.

We also recognize the limitations of all-atom MD simulations, and underscore that we do not attempt to use MD to address the large-scale shape change of chimallin that requires quaternary structural reorganization, i.e., extensive changes to the relative orientation of tetramers). Such large-scale changes clearly occur (in nature) on longer timescales (tens of milliseconds at least) than can be reasonably expected to be (spontaneously) reached with all-atom MD. Instead, to address some aspect of the larger-scale dynamics, we utilized elastic network models, which we posit do shed light on the key flexibility points (i.e., joints) along the sheet, and the coordinated movement of the domains relative to each other in the lattice.

Altogether, the computational and simulation-based analyses we present here augment the experimental data in a way currently inaccessible with experiment, providing a useful and unique perspective that we believe warrants inclusion in this paper.

L. 322-325. The prohead protease cleavage of chimallin is mentioned at the very end of the MS. However, this information must have been taken into account when the deletion mutants were designed (CTS1+2?). As mentioned above, in the in vivo competition experiment, the CTS1+2

mutant behaves abnormally compared to other linker mutants and this needs to be explained or at least commented on.

The truncation mutants were all designed to specifically disrupt particular interactions by precisely deleting these interacting segments, not based on the prohead protease cleavage site in phage PhiKZ.

The reviewer is correct that the behavior of the CTS1+2 truncation in the experiment shown in Figure 3d is somewhat inconsistent with the behavior of the other truncations. While its incorporation is reduced compared to full-length protein, this reduction is not statistically significant as with the other truncations studied. We do not have a clear explanation of this finding. As the reviewer suggests, we now specifically comment on this in the main text (Ln 176).

Finally, it would be interesting to learn if any of the chimallin deletion mutants affect phage morphogenesis. Assuming that chimallin is not integrated into the phage particle, i.e. the infected cell and phage progeny are unchanged except for the presence of chimallin mutants in addition to the WT in the cell, the mutants would be expected to lower the phage titer. Can this be correlated to the fraction of the mutant chimallin incorporated into the nucleus shown in Fig. 3d?

Please see our response to Reviewer #3's question 14. We have measured phage infectivity in the presence of a representative subset of ChmA mutants, and also measured phage growth curves in the presence of all ChmA truncations. The effects of these mutants on phage propagation is minimal, as detailed both above and in the new Supplementary Information Figure S2.

Reviewer Reports on the First Revision:

Referees' comments:

Referee #2 (Remarks to the Author):

The authors have addressed the concerns on molecular simulations and cryo-EM. The manuscript is now clear from the computational point of view, especially for what concerns pore volume, which was not completely clear in the previous version. The manuscript improved and I support the publication.

Referee #3 (Remarks to the Author):

I thank the authors for the detailed response. In particular the authors have added useful information about growth curves and the experimental design. They have adequately addressed my comments. I am convinced that this work will be of interest to many

Referee #4 (Remarks to the Author):

In the updated version of the MS "Architecture and self-assembly of the jumbo bacteriophage nuclear shell" the authors carefully addressed all reviewers' concerns - small and big - and this is much appreciated.

My major sticking point was the (originally perceived) lack of biological and functional novelty of the work because the putative identity of the protein, which forms the pseudo-nucleus, had been established in earlier studies and the square mesh of the nucleus wall had been visualized previously (albeit for a different phage).

However, as the authors correctly point out, the structure of the shell was unknown until now and the fact that it is made primarily out of the chimallin protein was also unknown. Furthermore, square mesh structures were previously found in the lysates of infected cells, so their relationship to the phage-encoded pseudo-nucleus was unknown. Here, the nucleus and its mesh-like wall was visualized for the first time in situ leaving no doubt about the structure of the wall.

My other comments and concerns were minor compared to the above. The authors dealt with them marvelously.

In conclusion, the study represents a real tour de force of a multi-scale and multi-approach structural characterization of a novel supra-molecular assembly. The study sets a very high bar for other researchers that are working on the characterization of novel cellular organelles and multicomponent assemblies.

Author Rebuttals to First Revision:

Referees' comments:

Referee #2 (Remarks to the Author):

The authors have addressed the concerns on molecular simulations and cryo-EM. The manuscript is now clear from the computational point of view, especially for what concerns pore volume, which was not completely clear in the previous version. The manuscript improved and I support the publication.

We thank the reviewer for their insights and for recommending our manuscript for publication.

Referee #3 (Remarks to the Author):

I thank the authors for the detailed response. In particular the authors have added useful information about growth curves and the experimental design. They have adequately addressed my comments. I am convinced that this work will be of interest to many

We thank the reviewer for their constructive feedback and support of our manuscript for publication.

Referee #4 (Remarks to the Author):

In the updated version of the MS "Architecture and self-assembly of the jumbo bacteriophage nuclear shell" the authors carefully addressed all reviewers' concerns - small and big - and this is much appreciated.

My major sticking point was the (originally perceived) lack of biological and functional novelty of the work because the putative identity of the protein, which forms the pseudo-nucleus, had been established in earlier studies and the square mesh of the nucleus wall had been visualized previously (albeit for a different phage).

However, as the authors correctly point out, the structure of the shell was unknown until now and the fact that it is made primarily out of the chimallin protein was also unknown. Furthermore, square mesh structures were previously found in the lysates of infected cells, so their relationship to the phage-encoded pseudo-nucleus was unknown. Here, the nucleus and its mesh-like wall was visualized for the first time in situ leaving no doubt about the structure of the wall.

My other comments and concerns were minor compared to the above. The authors dealt with them marvelously.

In conclusion, the study represents a real tour de force of a multi-scale and multi-approach structural characterization of a novel supra-molecular assembly. The study sets a very high bar for other researchers that are working on the characterization of novel cellular organelles and multicomponent assemblies.

We thank the reviewer for their questions, and we are glad that our replies were effective in addressing their concerns. We are grateful for the enthusiasm of their recommendation.